# Transcriptomic profiling of lung alveolar macrophages reveals distinct contribution of sterol metabolism in macrophage response to *Cryptococcus gattii* infection

Siranart Jeerawattanawart[1,2], Adithap Hansakon[3], Rudi Alberts[4,5], Yongliang Zhang[4,5], Pornpimon Angkasekwinai [2,6]*

**1** Faculty of Medical Technology, Rangsit University, Pathum Thani, Thailand, **2** Department of Medical Technology, Faculty of Allied Health Sciences, Thammasat University, Pathum Thani Thailand, **3** Chulabhorn International College of Medicine, Thammasat University, Pathum Thani Thailand, **4** Department of Microbiology and Immunology, NUSMED TRP Immunology, Yong Loo Lin School of Medicine, Singapore, Singapore, **5** Immunology Programme, Life Science Institute, National University of Singapore, Singapore, Singapore, **6** Research Unit in Molecular Pathogenesis and Immunology of Infectious Diseases, Thammasat University, Pathum Thani, Thailand

* upornpim@tu.ac.th, p.akswn@gmail.com

## Abstract

Alveolar macrophages are well known as the first responders to pulmonary cryptococcosis; however, their dynamic molecular responses to *Cryptococcus gattii* infection in vivo remain limited. Here, we investigated the transcriptional profiles of lung alveolar macrophages purified from mice intranasally infected with *C. gattii* and compared them to those infected with *C. neoformans* using RNA sequencing analysis. Alveolar macrophages from *C. gattii*-infected mice exhibited distinct transcriptional alterations, particularly in genes associated with sterol biosynthesis, whereas those from *C. neoformans*-infected mice showed enrichment in interferon and cytokine signaling pathways. Treatment with the sterol biosynthesis inhibitor lovastatin significantly reduced cholesterol accumulation in *C. gattii*-infected RAW 264.7 cells. Furthermore, lovastatin treatment of *C. gattii*-infected RAW 264.7 and bone marrow-derived macrophages suppressed intracellular cryptococcal proliferation and augmented inflammatory response, as evidenced by increased expression of *Nos2* and *Il6*. Lovastatin also potentiated the antifungal effect of fluconazole by promoting intracellular fungal clearance and increasing nitric oxide activity in macrophages. In *C. gattii*-infected mice, lovastatin treatment enhanced the efficacy of fluconazole, resulting in improved pulmonary fungal clearance, increased lung CD4+T cell numbers, and elevated nitric oxide activity. Collectively, these findings reveal a unique macrophage response to *C. gattii* infection and highlight the role of sterol metabolism in modulating host defense, offering potential avenues for therapeutic intervention.

**Data availability statement:** All relevant data are within the paper and its Supporting Information files.

**Funding:** This study was supported by the Thailand Science Research and Innovation (TSRI) Fundamental Fund fiscal year 2024, Thammasat University and the Thammasat University Research Unit in Molecular Pathogenesis and Immunology of Infectious Diseases. The funders had no role in study design, data collection and analysis, decision to publish, or preparation of the manuscript.

**Competing interests:** The authors declared no conflicting financial interests.

## Introduction

In late 2022, the World Health Organization (WHO) announced the fungal priority pathogens list (FPPL) to emphasize significant fungal infections causing worldwide health-threatening impact [1,2]. *Cryptococcus neoformans* and *Cryptococcus gattii* have been classified as a critical-priority fungal pathogen and a medium-priority fungal pathogen, respectively. Different species of *Cryptococcus* displayed distinct differences in epidemiology, clinical features, and virulence [3,4]. Both *C. neoformans* and *C. gattii* were globally spreading and life-threatening fungal pathogens that caused disease in immunocompromised and immunocompetent patients, respectively. A hallmark of both species is the presence of a polysaccharide capsule composed of glucuronoxylomannan (GXM) and glucuronoxylomanogalactan (GXMGal). These capsule components are critical virulence factors that facilitate immune evasion, inhibit phagocytosis, and modulate host immune responses [5,6]. Clinically, *C. gattii* infection is more commonly associated with pulmonary cryptococcosis, whereas *C. neoformans* infection typically presents as cryptococcal meningitis [7].

Upon infection, a heterogenous population of macrophages was found to be regulated in the lungs. Pulmonary macrophages are primarily classified into alveolar macrophages (AMs) and interstitial macrophages (IMs), which refer to their specific locations within the lung alveolar cavity and interstitial tissue, respectively [8–10]. More recent study using flow cytometry observed four subsets of macrophages in the lung including AMs, IMs, Ly6c+, and Ly6c- monocyte-like macrophages after *C. neoformans* infection [11]. Among them, alveolar macrophages are recognized as the first responders to *Cryptococcus* infection in the lungs [12]. After inhalation of spores or desiccated yeast into alveolar cavities, AMs play important roles by mediating fungal uptake and facilitating fungal clearance or fungal dissemination [13,14], depending on host cytokine factor [15,16] and virulence of the cryptococcal strains [17]. The cytokine microenvironment plays a critical role in driving macrophage polarization into either the M1 or M2 phenotype [18]. M1 macrophage activation requires stimulation by IFN-γ, leading to fungal clearance through the induction of reactive oxygen species (ROS) and reactive nitrogen species (RNS). In contrast, IL-4 and IL-13 cytokines promote the polarization of M2 macrophages, which facilitates intracellular fungal proliferation and survival [19].

Several studies have investigated the differences in host responses, particularly in macrophages, during *C. neoformans* and *C. gattii* infections [20–22]. However, the regulation by these fungi may differ among various types of macrophages. Recent studies have underscored the role of TLR9 expressed in brain microglial cells during *C. gattii* infection. In TLR9-deficient mice, the absence of this receptor may facilitate the entry of yeast cells into the brain [23,24]. Transcriptomic analysis of IFN-γ-pretreated J774 macrophage cells infected with *C. neoformans* H99 and *C. gattii* R265 revealed potential involvement of AKT/mTOR pathway in promoting fungal survival [21]. However, bone marrow-derived macrophages (BMDMs) showed no significant differences in the transcriptomic response when infected with either *C. neoformans* H99 or *C. gattii* R265 [20]. These findings highlight the need for further investigation using in vivo mouse models or primary human-derived

macrophages to better understand host responses to these two *Cryptococcus* species. More recently, a transcriptomic analysis of peripheral blood mononuclear cell (PBMC) isolated from mice infected with *C. neoformans* H99 showed greater involvement of immune response-related genes, including those associated with Th2 and complement pathway, than those infected with *C. gattii* R265 [22]. Consequently, the transcriptomic profile of alveolar macrophages from mice infected with *C. neoformans* H99 indicated a shift in gene expression associated with M1 to M2 macrophage polarization during disease progression from days 7–14 [25]. Although alveolar macrophages are the first line of defense against respiratory pathogens and known to be associated with disease pathogenesis caused by *C. neoformans* infection, the knowledge of the underlying mechanism in response to *C. gattii* R265 remains limited.

In this study, we aimed to investigate the transcriptional profiles of alveolar macrophages isolated from mice intranasally infected with *C. gattii* R265 using RNA sequencing, with the goal of identifying key molecular pathways involved in the host immune response. Based on the observed upregulation of sterol biosynthesis-related genes in alveolar macrophages following *C. gattii* infection, we further sought to examine the effects of sterol biosynthesis inhibition through lovastatin treatment, both alone and in combination with the anti-fungal drug fluconazole, on macrophage responses in vitro. Additionally, we evaluated the effect of these treatments in a mouse model of pulmonary cryptococcosis. These investigations may advance our understanding of host-pathogen interactions and support the development of potential therapeutic strategies for *C. gattii* infections.

## Materials and methods

### *Cryptococcus* strains and culture conditions

*C. gattii* (R265) and *C. neoformans* (H99) strains were obtained from the Department of Medical Technology, Faculty of Allied Health Sciences, Thammasat University. All strains were grown in Sabouraud dextrose agar (SDA) (HiMedia) at 25°C for 48 hours. After incubation, a single colony of yeast cells was cultivated in Sabouraud dextrose broth (SDB) (BD Difco™) at 37°C with agitating at 200 rpm for 24 hours. Then yeast cells were harvested, centrifuged at 3,000 rpm for 5 minutes, gently washed thrice with sterile Dulbecco's phosphate-buffered saline (DPBS), counted using a hemacytometer, and adjusted the yeast cell concentration in phosphate-buffered saline (PBS) before subsequent infection.

### Animals

The age-(6- to 8- weeks old) and gender-matched BALB/c mice were utilized in all experimental studies. All mice were purchased from Nomura Siam International Co., Ltd., Thailand, and housed under specific pathogen-free conditions (a 12-hours light/dark cycle, temperature of $22 \pm 1$ °C, and humidity of 30–70%) with access to food and water ad libitum in the animal facility of Thammasat University. Mice were allowed to acclimate for 1 week before undergoing any experimental procedures. All animal procedures were performed according to the relevant ethical ARRIVE guidelines (https://arriveguidelines.org) and approved by the Thammasat University Animal Care and Use Committee (Protocol Number: 016/2023).

### In vivo *Cryptococcus* infection

The mouse model of *Cryptococcus* infection was performed as previously reported [26]. Briefly, BALB/c mice (6−10 mice per group: for alveolar macrophage isolation) were anesthetized using 3% isoflurane delivered via a vaporizer with an oxygen flow rate of 0.9 L/min through an induction chamber prior to the intranasal administration of PBS, *C. gattii*, and *C. neoformans* cell suspension at $5 \times 10^4$ cells/mouse. To investigate the effect of lovastatin on host response to *C. gattii* infection, infected mice were divided into four groups (n = 3 mice per group): vehicle control, 4 mg/kg lovastatin [27], 20 mg/kg fluconazole [28], and 4 mg/kg lovastatin plus 20 mg/kg fluconazole. Lovastatin was dissolved in a solution containing 40% polyethylene glycol 300 (PEG 300), 5% Tween 80, 45% saline, and 10% dimethyl sulfoxide (DMSO). Fluconazole

was prepared in sterile PBS (pH 7.2–7.4). All treatments were injected intraperitoneally once daily for six consecutive days after infection. Mice were monitored daily for health status and behavioral changes. In cases where the mice exhibited abnormalities such as immobility, loss of appetite, or other observable signs of distress or dysfunction, humane endpoints were applied. At 7 days post infection, mice were sacrificed by euthanasia via controlled gradual displacement with carbon dioxide using a flow meter in accordance with IACUC and the American Veterinary Medicine Association (AMVA) guidelines on euthanasia.

### Lung fungal burden assessment

Lung samples were collected and homogenized in sterile PBS containing 1% penicillin-streptomycin. Following homogenization, lung homogenates were subjected to serial dilution and plated on SDA. To evaluate fungal burden, fungal colony-forming units (CFUs) were subsequently quantified after 48 hours of incubation at 30°C and expressed as Log CFU per gram of lung tissue (Log CFU/g).

### Isolation and sorting of alveolar macrophages

AMs were isolated and prepared as previously described [29]. Briefly, lung tissues were collected from mice treated with PBS or infected with *C. gattii* and *C. neoformans* (6–10 mice per group). The tissues were finely minced through 41 µM-sterile nylon mesh (Merck) in complete Roswell Park memorial institute (RPMI) 1640 containing 10% heat-inactivated FBS, 2 mM L-glutamine, and 100 U/mL penicillin-streptomycin. After centrifugation, the cell pellet was resuspended in ACK lysis buffer for red blood cell lysis. The remaining cells were reconstituted in 20% Percoll (GE Healthcare) and centrifuged at 3,000 rpm for 25 minutes at 25°C. The upper layer was discarded, and the cell pellet containing total lung leukocyte was resuspended with complete RPMI medium and counted before further experiments.

To isolate alveolar macrophages, total lung leukocytes were blocked with anti-CD16 and stained with fluorochrome-conjugated antibodies: fluorescein isothiocyanate (FITC)-conjugated anti-CD11b (M1/70, BD Pharmingen, San Jose, CA), Phycoerythrin (PE)-conjugated anti-Siglec F (E50-2440, BioLegend), allophycocyanin (APC)-conjugated anti-Gr.1 (RB6-8C5, BioLegend), and PerCP-Cy5.5-conjugated anti-CD11c (N418, BioLegend) in the dark at 4°C for 30 minutes. After washing with FACS staining buffer, the stained cells were then stained with 7-amino actinomycin D (7-AAD) to remove debris and dead cells. FACSAria III was used to sort the population of alveolar macrophages.

### RNA sample preparation and RNA sequencing

Isolated alveolar macrophages from PBS-treated or *C. gattii*-infected and *C. neoformans*-infected mice from three independent experiments were extracted for total RNA using a RNeasy plus mini kit (Qiagen) according to the manufacturer's instruction. The total RNA yield was pooled and measured using a Nanodrop™ One Spectrophotometer (Thermo Scientific, USA). Total RNA at 600 ng was utilized for cDNA library preparation and sequencing at Novogene Corporation Inc. Briefly, the paired-end cDNA library was constructed using the NEBNext Ultra RNA library prep kit for Illumina (New England BioLabs, Ipswich, MA), with polyA enrichment conducted via Oligo(dT) beads, followed by fragmentation into small pieces using divalent cations. Double-stranded cDNA (dscDNA) was generated using random primers. After the dscDNA was modified by converting into blunt end, adding adenylate 3′ ends, NEBNext adaptor ligation, and size selection, the dscDNA libraries were amplified by PCR and purified the PCR product using AMPure XP system. The purified dscDNA libraries were determined for the concentration and quality using a Qubit 2.0 fluorometer and Agilent 2100 bioanalyzer to ensure its quality with qPCR library quantification kit. Lastly, the purified dscDNA libraries were sequenced on a high-throughput Illumina HiSeq PE150 platform. The raw sequence read was assessed for quality using an in-house Perl script quality control, which is demonstrated by the per-base quality Phred Q score30, sequencing error rate distribution, and per-read GC content (S1 Table). Particularly, the high-quality reads (clean reads) in the dataset were included for bioinformatics analysis. The RNA sequencing data were provided in the National Center for Biotechnology Information Gene

Expression Omnibus database repository under the accession number GSE281113 (https://www.ncbi.nlm.nih.gov/geo/query/acc.cgi?&acc=GSE281113).

## RNA sequencing data analysis

For reads mapping and gene expression quantification, raw sequence reads were aligned to the mouse genome (*Mus musculus*, GRCm38) using TopHat2 v2.0.12/Bowtie v2.2.3. Read data were annotated using NCBI BLAST v2.2.28 based on known mouse genes with an expected value of $1.0 \times 10^{-10}$. To estimate gene expression levels, HTSeq v0.6.1.was used to count the read numbers mapped of each gene and calculated the fragments per kilobase of transcript per million (FPKM) of base pairs sequenced which is a normalized measure of relative abundance of transcripts according to the length and reads count mapped of the gene.

For differential gene expression analysis, the read data were analyzed using the Bioconductor EdgeR software through one scaling normalized factor. Thereafter, the DEGs were conducted by the R package DESeq2 v1.18.1 using the Benjamini and Hochberg method for multiple testing $p$ value correction. Following optimizing for the false discovery rate, significant differentially expressed genes were identified as $p$ value <0.05. A fold change >2 log2 between the compared groups was indicated as upregulation genes, while a fold change < −2 log2 between the compared groups was indicated as downregulation genes. A pairwise comparison of DEGs was carried out, which included (1) AMs of the *C. gattii* R265-infected mice at 7 days after infection compared with those of the PBS-treated mice and (2) AMs of the *C. neoformans* H99-infected mice at 7 days after infection compared with those of the PBS-treated mice. A Venn diagram was created by BioinfoGP Venny v2.1.0 software to illustrate the number of differentially expressed genes unique to each group and the overlaps shared between groups.

## Gene ontology (GO) enrichment analysis

GO analysis was performed to determine the regulatory roles of these distinctively expressed DEGs of AMs compared between different groups using GO Seq (v1.34.1). The significance of GO term enrichment was indicated by the FDR-corrected $p$ value of <0.05.

## Analysis of protein-protein interaction network

To understand the possible pathways associated with the DEGs, a protein-protein interaction (PPI) network was established by applying the search tool to retrieve the interactions among genes/protein (STRING) database. Cytoscape3 v2.0.2 software was used to create the network of interactions.

## Bronchoalveolar lavage collection and enrichment of AMs

As previously reported [25], bronchoalveolar lavage (BAL) was taken from PBS-treated mice or *C. gattii*-infected and *C. neoformans*-infected mice. Briefly, the trachea was opened by making surgical incision under the jaw. A 22-gauge catheter was introduced into the airway and held in place with a string. Ice-cold PBS containing 5 mM EDTA was administered through the catheter to a volume of 0.5 mL and the lung was then washed 5 times before collecting the BAL suspension into the new 1.5 mL tube. The BAL suspension was centrifuged at 3,000 rpm for 5 minutes at 25°C. After discarding supernatant, the BAL cells were cultured in serum-free medium for 2 hours in a 24-well plate at 37°C, 5% $CO_2$ for enrichment of AMs. After 2 hours of incubation, the cells were gently washed with DPBS, and the adherent cells containing AMs were harvested in TRIzol® reagent (Invitrogen) and stored at −20°C until used for gene expression analysis.

## Validation of the data by using quantitative real-time PCR

The RNA samples were isolated using TRIzol® reagent according to the manufacturer's instructions. Then, cDNA was synthesized from total RNA at 0.1 ng to 5 μg using oligodT and RiboLock RNase Inhibitor (Thermo Scientific™), and

RevertAid Reverse Transcriptase (Thermo Scientific™). These cDNA samples were amplified in iTaq™ universal SYRB® Green Supermix Biorad Laboratories) with the primers included in S2 Table using CFX96 Touch Real-Time PCR Detection System (Biorad Laboratories) to quantify the target gene expression. Endogenous actin (*Actb*) transcript levels served as an internal control for the normalization of target gene expression.

## RAW 264.7 cell culture

The murine macrophage cell line RAW 264.7 (ATCC TIB-71) was cultured in high-glucose Dulbecco' modified Eagle medium (DMEM) (HyClone™), supplemented with 10% heat-inactivated fetal bovine serum (FBS, Gibco™), 1% penicillin-streptomycin (P/S), and 1% L-glutamine. The cells were maintained at 37°C in a humidified chamber of 5% $CO_2$ and subcultured 2–3 times weekly. Cells at 85–90% confluence were used for further experiments.

## Preparation of BMDMs

BMDMs were prepared as previously reported with some modification [25]. Briefly, tibias and femurs were obtained from BALB/c mice and then flushed with ice-cold complete high-glucose DMEM. The collected cells were centrifuged at 1,500 rpm for 5 minutes at 25°C. After discarding supernatant, the cell pellet was immediately resuspended with ACK buffer for RBC lysis. Following centrifugation, these cells were resuspended in complete DMEM and filtered through 41 μM-sterile nylon mesh. The remaining cells were counted and seeded at 20 x $10^6$ cells in 60 x 15 mm cell culture dish before being incubated at 37°C in 5% $CO_2$ for 4 hours. After incubation, non-adherent cells were harvested and induced macrophage differentiation in complete DMEM containing 20 ng/mL M-CSF (PeproTech) and cultured at 37°C with 5% $CO_2$. After 3 days, cells were replaced with fresh complete DMEM medium containing 20 ng/mL M-CSF. The phenotype of the BMDMs was confirmed by staining for CD11b. Experiments were performed using differentiated BMDMs for 6 days.

## In vitro treatment of *Cryptococcus*-infected macrophages with Lovastatin

The effect of lovastatin on in vitro phagocytosis, intracellular cryptococcal load and intracellular proliferation rate in macrophages infected with *Cryptococcus* were performed according to previously described [25]. Briefly, RAW 264.7 cells or BMDMs were collected and plated at 2 x $10^5$ cells/well in 24 well plate at 37°C in 5% $CO_2$ for 24 hours. Cells were then cultured in serum-free DMEM for 2 hours and stimulated with 1 μg/mL PMA (Sigma-Aldrich) for 30 minutes before infection. For determining phagocytosis, activated macrophage cells were infected with *C. gattii* R265 that had been opsonized with 1 mg/mL anti-capsule mAb 18B7 (Merck). The infection was performed at a density of 2 x $10^6$ cells/well (MOI 10) in the presence of vehicle control DMSO or lovastatin (10 μM) [30] for 2 hours. After that, cells were washed with sterile DPBS to remove non-phagocytosed yeast cells. Macrophages were lysed with sterile distilled water at 37°C for 20 minutes before plating on SDA for CFU counting. To investigate the direct effect of lovastatin on *Cryptococcus*-infected macrophages, lovastatin (10 μM) was added into *Cryptococcus*-infected macrophages after the removal of non-phagocytosed yeast cells at 2 hours. The intracellular cryptococcal load (ICL) was determined at 24 hours by lysing the cells and plating on SDA, as previously described, and the intracellular proliferation rate (IPR) of fungi was calculated by dividing the number of intracellular cryptococci at 24 hpi by the number of intracellular crypto-cocci uptake at 2 hpi. In some experiments, to investigate whether lovastatin could enhance the efficacy of fluconazole treatment in *C. gattii* R265-infected macrophages, fluconazole (20 μg/mL) was added with or without lovastatin (10 μM) after the removal of non-phagocytosed yeast. The concentration of fluconazole (20 μg/mL) was selected based on a previous study to examine its effect on BMDMs in response to *C. gattii* [28]. Then ICL and IPR were performed at 24 hpi, and the number of extracellular yeasts also enumerated in the culture supernatant from each condition by plat-ing on SDA.

## Intracellular cholesterol measurement

To measure cholesterol levels within macrophages, intracellular cholesterol assay was performed as previously described with some modification [31,32]. Briefly, after 24 hours of infection, RAW 264.7 cells from various treatment groups were washed twice with sterile cold DPBS. After washing, cells were homogenized in hexane: isopropanol solution (3:2, v/v) and centrifuged at 15,000 x g at 4°C for 10 minutes. The cell lysates (organic phase) were harvested and vacuum-dried under 45°C for 30 minutes to remove the chloroform. The vacuum-dried samples were dissolved in isopropanol: Nonidet P-40 (NP40) (9:1, v/v) via pipetting and vortexing. The total cholesterol content in cells was measured using the Stanbio Cholesterol LiquiColor™ (Stanbio Laboratory) and normalized to the total protein concentration of each sample using the Pierce™ BCA Protein Assay Kit (Thermo Fisher Scientific).

## IL-6 ELISA assay

Culture supernatants from BMDMs under various conditions were collected and stored at −80°C until use. IL-6 levels were measured using ELISA kits (PeproTech) according to the manufacturer's instructions.

## ROS quantification using flow cytometry

To measure the ROS production of BMDMs, ROS assay was conducted as previously described with some modifications [33]. Following 24 hours of infection, BMDMs from different treatment groups were washed with sterile DPBS, collected after centrifugation, and resuspended in Zombie Yellow dye (BioLegend) for cell viability staining. After washing, the cells were blocked with anti-CD16/CD32 (Biolegend) and stained for the cell surface expression of CD11b with PerCP-Cy5.5– conjugated anti-CD11b (M1/70; BioLegend). To assess ROS production, the stained cells were treated with 100 µL of 10 µM H2DCF-DA in 2% BSA solution (Sigma-Aldrich) for 30 minutes at 37°C. Flow cytometry analysis was performed using a BD FACSLyric cytometer (BD Biosciences), and the collected data were analyzed with FlowJo™ Software (Tree Star, Inc.).

## Nitric oxide measurement

Culture supernatants of BMDMs with different treatments or BAL fluid were used to detect nitric oxide using Griess reagent (1% sulfanilamide, 0.1% N-1-napthylethylenediamine, 5% orthophosphoric acid, Sigma Aldrich). Nitrite concentration was determined by measuring absorbance at 570 nm and calculating it to a standard curve (ranging from 0 to 500 mM).

## Analysis of lung inflammatory cell infiltration by flow cytometry

Bronchoalveolar lavage (BAL) fluid was collected by flushing the lungs of mice five times with 500 µL ice-cold PBS containing 5 mM EDTA through an intratracheal catheter, followed by gentle aspiration. The recovered BAL fluid was centrifuged at 3,000 rpm for 10 minutes, and the cell pellet was counted using trypan blue. Cells were resuspended in Zombie Yellow dye (BioLegend) for viability staining and subsequently stained with a panel of fluorochrome-conjugated antibodies to analyze lung inflammatory cell populations. The antibody panel included FITC-conjugated anti-CD11b clone M1/70 (BD Pharmingen), PE-conjugated anti-Siglec-F clone E50-2440 (BioLegend), PerCP-Cy5.5-conjugated anti-CD11c clone HL3 (BioLegend), allophycocyanin-conjugated anti-Gr-1 clone RB6-8C5 (BioLegend), V500-conjugated anti-CD4 clone GK1.5 (eBioscience), and PE-Cy7-conjugated anti-CD3e clone 145-2C11 (BD Pharmingen). To prevent nonspecific binding, Fc receptors were blocked with an anti-CD16/CD32 antibody (BioLegend). Debris and dead cells were excluded by gating based on forward and side scatter profiles and viability staining. Inflammatory cell populations were classified as CD4+T cells (CD3+ CD4+), dendritic cells (DC; CD11b+CD11c+), alveolar macrophages (CD11c+ Siglec-F+), eosinophils (CD11c- Siglec-F+), and neutrophils (CD11b+ Gr.1hi). Flow cytometry was performed using a BD FACSLyric cytometer (BD Biosciences), and data were analyzed with FlowJo™ software (Tree Star).

## Statistical analysis

Experiments were carried out at least two to three times. All data were expressed as mean values ± SD. Data were analyzed using the unpaired t test (two-tailed) and one-way ANOVA with Turkey's post hoc analysis. Statistical analyses were performed using GraphPad Prism 10 software. A *p* value <0.05 was considered statistically significant.

## Results

### Transcriptomic profiling of alveolar macrophages infected with *C. gattii* and *C. neoformans*

To determine the effect of *C. gattii* and *C. neoformans* infection on alveolar macrophage response, BALB/c mice were intranasally treated with PBS or infected with *C. gattii* (R265) or with *C. neoformans* (H99). At 7 days postinfection, alveolar macrophages identified as SiglecF^hiCD11c^hiCD11b^lo were isolated from their lungs using fluorescence-activated cell sorting. Sorted cells were pooled to obtain enough RNA quantity and used to further subject for RNA-sequencing (Fig 1A). A raw dataset consisting of approximate 40 million reads (~6 Gbps) was yielded. After filtering low-quality reads, adaptor or ambiguous sequences, and removal of contamination, 34.51 (86.52%) million clean reads from alveolar macrophages purified from the PBS-treated mice, 35.31 (88.14%) million clean reads from those purified from the *C. gattii*-infected mice, and 41.61 (91.82%) million clean reads from those purified from the *C. neoformans*-infected mice were retained. The average of Q30 of clean reads was > 85%, indicating that the obtained clean reads were of high quality, and the GC contents of the clean data for all samples ranged between 49.67 to 50.96% (S1 Table).

Approximately 91.87 to 92.81% of clean reads were successfully mapped to the reference *Mus musculus* genome. A Venn diagram showed the pairwise comparison of significantly differentially expressed genes (DEGs) from alveolar macrophages of mice infected with *C. gattii* and *C. neoformans* relative to those treated with PBS. A total of 540 significant DEGs (*p* < 0.05) were unique to AMs of *C. neoformans*-infected mice, while only 51 DEGs were unique to the *C. gattii*-infected mice. Additionally, 115 DEGs were found to be common in both groups (Fig 1B). Compared to the alveolar macrophages of PBS-treated mice, those from *C. neoformans*-infected mice displayed significant upregulation of genes related to interferon β signaling (*Ligp1*, *Tgtp2*, *Gm4951*) and cellular response to cytokine (*Nos2*), while the most downregulated genes were associated with to oxygen binding (*Hbb-bs*, *Hba-a2*, *Hba-a1*, *Alas2*) (Fig 1B and S3 Table). However, alveolar macrophages of mice infected with *C. gattii* showed notable upregulation of genes related to sterol biosynthesis (*Fdps, Cyp51, Sqle*) and downregulation of genes associated with lipid particle (*Plin1*) and cholesterol efflux (*Abca1*) (Fig 1B and S4 Table). Altogether, these transcriptome data revealed distinct changes in gene expression of alveolar macrophages upon infection with *C. gattii* and *C. neoformans*, suggesting differential macrophage response to *C. gattii* and *C. neoformans* infections.

### Gene ontology (GO) analysis reveals the altered pathway of macrophage sterol metabolism upon *C. gattii* infection

Because alveolar macrophages in response to *C. gattii* infection revealed unique gene expression changes, we further investigated the alterations in the macrophage transcriptome in reacting to infections caused by *C. gattii* using the gene ontology functional analysis. The GO enrichment analysis showed that the most significantly upregulated DEGs were mainly related to sterol biosynthesis and metabolic processes. The upregulated genes were also engaged in other pathways, including the organic hydroxy compound biosynthetic process, the small molecule biosynthetic process, and leukocyte chemotaxis (Fig 2A and S5 Table). Surprisingly, the most significantly downregulated DEGs in macrophage response to *C. gattii* were related to the regulation of plasma lipoprotein, basolateral plasma membrane, positive regulation of cholesterol efflux and lipid particle (Fig 2A and S6 Table). Using string analysis to investigate the interaction network of upregulated DEGs belonging to the top GO pathway analysis in AMs upon *C. gattii* infection, we observed an increase in genes associated with sterol biosynthesis and metabolic processes (*Fdps*, *Cyp51*, *Sqle*, *Sc5d*, *Hsd17b7*, *Fdft1*, *Ldlr*, *Msmo1*, *Ch25h*, *Dhcr24*, *Hmgcs1*, *Lss*, *Hmgcr*) in response to *C. gattii* infection (Fig 2B). In contrast, the interaction networks of

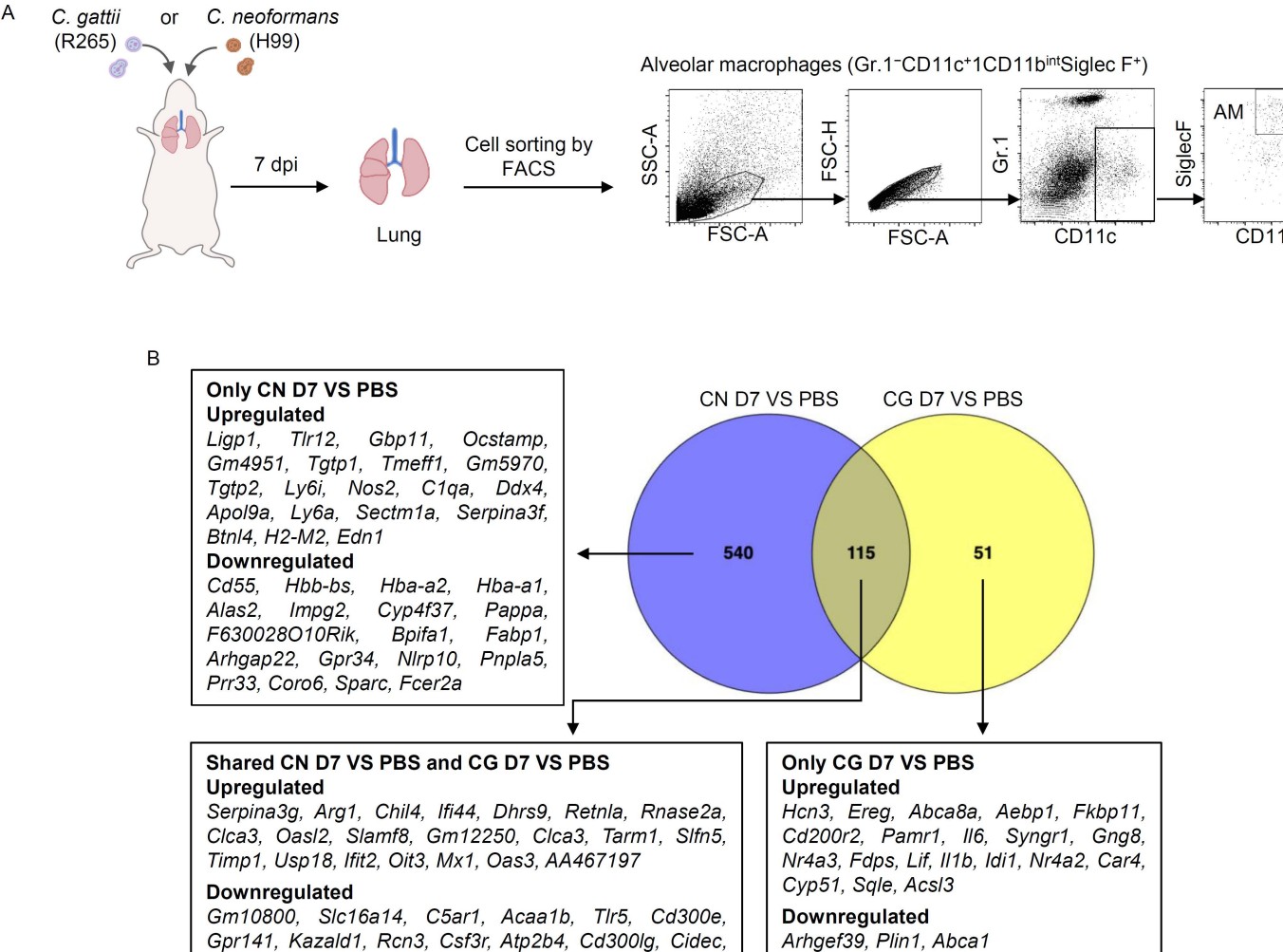

**Fig 1. Transcriptome profiles of alveolar macrophages in response to *C. gattii* and *C. neoformans* infection.** (A) BALB/c mice were intranasally infected with *C. gattii* R265, *C. neoformans* H99, or treated with PBS. At 7 days post-infection, alveolar macrophages (AMs) were isolated from the lungs using fluorescence-activated cell sorting. Samples from 6–10 mice per group were pooled across three independent experiments for transcriptome analysis. Representative plots show the gating strategy for analyzing and sorting AMs (CD11c+CD11b$^{int}$Siglec F+) from the lung. (B) Venn diagrams compared the significant DEGs in lung AMs infected with *C. gattii* at 7 dpi (CG D7) versus PBS and lung AMs infected with *C. neoformans* at 7 dpi (CN D7) versus PBS. The overlapping sections of the Venn diagrams represent the common DEGs shared between the compared transcriptomes, while the non-overlapping sections represent DEGs unique to each strain when compared to PBS. The top upregulated and downregulated DEGs that are unique or shared in each comparison are displayed.

downregulated DEGs from the top GO pathways analysis in AMs during *C. gattii* infection revealed the reduction of genes related to the regulation of plasma lipoprotein (*Mylip*, *Abca1*, *Pltp*), positive regulation of cholesterol efflux (*Abca1*, *Pltp*), and lipid particle (*Plin1*, *Cidec*) in alveolar macrophages of mice infected with *C. gattii* (Fig 2B). We conducted additional gene expression analysis in alveolar macrophages to confirm the alteration of host lipid metabolism. The selected genes from both upregulated and downregulated DEGs related to lipid metabolism were analyzed using quantitative real-time PCR. In accordance with the DEGs results, *C. gattii* infection markedly elevated the expression of genes linked to sterol biosynthesis, including *Fdps*, *Cyp51*, *Sqle*, *Hsd17b7*, and *Lss*, but not *Sc5d* and *Fdtf1* (Fig 2C). However, the expression

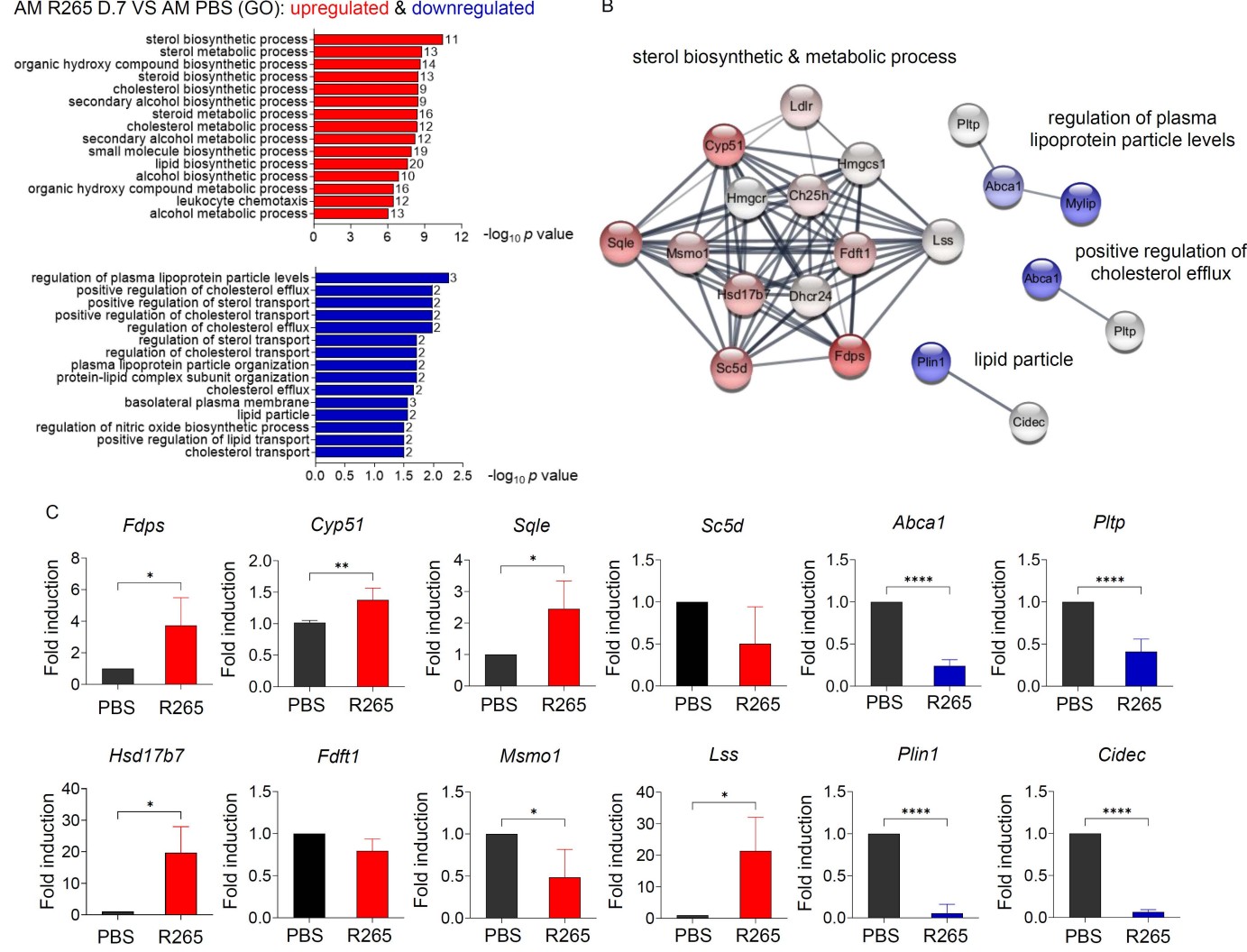

**Fig 2. Gene ontology and network analysis of alveolar macrophage response to *C. gattii* infection.** (A) Gene Ontology analysis revealed the functional annotation of the top 15 significant pathways for the upregulated DEGs (red bars) and downregulated DEGs (blue bars) in comparison between lung AMs of mice infected with *C. gattii* and those treated with PBS. (B) The interaction network of upregulated and downregulated DEGs from the GO enrichment analysis in AMs, comparing those from *C. gattii*-infected mice with those from PBS-treated control. The expression levels of DEGs are presented by the intensity of red for upregulated genes and blue for downregulated genes. BALB/c mice were intranasally infected with *C. gattii* R265. At 7 days post-infection, AMs were isolated, and the expression of selected genes was measured using quantitative real-time PCR. (C) Quantitative real-time PCR analysis of sterol biosynthesis-associated genes (*Fdps*, *Cyp51*, *Sqle*, *Sc5d*, *Hsd17b7*, *Fdft1*, *Msmo1*, *Lss*), cholesterol efflux-associated genes (*Abca1*, *Pltp*) and lipid particle-associated genes (*Plin1*, *Cidec*). Data are expressed as fold induction over actin (*Actb*) expression, with the mRNA levels in AMs of PBS-treated mice normalized to 1 and compared to those from mice infected with R265 for 7 days. Graphs depict mean ± SD of three independent experiments. Significance was determined using Student's t-test analysis (*$p < 0.05$, **$p < 0.01$, ****$p < 0.0001$).

of *Msmo1*, which acts downstream of *Cyp51* enzyme by catalyzing the demethylation of methylsterols to support cholesterol production [34], was downregulated compared to other sterol biosynthesis genes (Fig 2C). This downregulation might be attributed to the saturation of the conversion step from lanosterol to zymosterone. We also observed a reduction in the expression of genes involved in cholesterol efflux (*Abca1* and *Pltp*) and lipid particle (*Plin1* and *Cidec*) (Fig 2C). These data indicated that *C. gattii* infection uniquely alters transcriptional patterns in alveolar macrophages by modifying host lipid metabolism.

## Modulation of sterol metabolism influences macrophage response to *C. gattii* infection

Since numerous studies showed that statin (HMG-CoA reductase inhibitors) could lower cholesterol levels in macrophages [28,35]. We further investigated whether disrupting cholesterol metabolism could influence *C. gattii* infection in macrophages (Fig 3A). Previous study has demonstrated that lovastatin at a concentration of 10 μM potentially suppressed cholesterol synthesis and reduced cholesterol levels in RAW 264.7 cells, without affecting cell viability [30]. In our study, we observed a similar reduction in cholesterol levels in these cells after treatment with lovastatin at the same concentration for 24 hours (Fig 3B). Furthermore, cholesterol levels were increased in *C. gattii*-infected RAW 264.7 cells, while a decrease in cholesterol content was observed after lovastatin treatment at 24 hpi (Fig 3B). Since previous studies have reported the antifungal properties of lovastatin in inhibiting the growth of *Mucor racemosus* [36] and *Candida albicans* [37,38], we tested the direct effect of this drug at concentration of 10 μM on the growth of *C. gattii* and compared it with fluconazole, as well as a combination of lovastatin and fluconazole. Unlike fluconazole, lovastatin at concentration of 10 μM neither inhibited the growth of *C. gattii* nor exhibited synergistic effect when combined with fluconazole, compared to the vehicle control treatment (S1 Fig). We further determined the effect of lovastatin on the phagocytic activity of RAW 264.7 cells against *C. gattii* R265 infection (Fig 3C). Indeed, RAW 264.7 macrophages treated with lovastatin during *C. gattii* R265 infection exhibited phagocytosis capabilities comparable to those of macrophages treated with DMSO (Fig 3D). In addition, we examined the effect of lovastatin on macrophage function in controlling intracellular fungal proliferation by treating infected macrophages with lovastatin after removing non-phagocytosed yeast cells at 2 hpi (Fig 3E). Our results showed that lovastatin treatment significantly decreased both the intracellular cryptococcal load (ICL) and intracellular proliferation rate (IPR) at 24 hpi in *C. gattii*-infected RAW 264.7 cells (Fig 3F). This effect was further confirmed in BMDMs, where lovastatin similarly decreased ICL and IPR following *C. gattii* infection (Fig 3G). Collectively, our data indicated that cholesterol-lowering drug lovastatin modulates the macrophage response to *C. gattii* infection by enhancing their antifungal activity.

## The effect of lovastatin on inflammatory response by macrophages during *C. gattii* infection

Given that alterations in cholesterol metabolism inside macrophages affected their effector functions and inflammatory responses [39,40], we subsequently assessed the mechanisms by which lovastatin treatment mitigated *C. gattii* R265 infection in macrophages. The effect of lovastatin treatment on the mRNA expression of proinflammatory markers (*Nos2*, *Il6*, *Il1b*, *Tnfa*, *Mmp9*, *Tlr4*), anti-inflammatory markers (*Arg1*, *Fizz1*, *Ym1*) and lysosome-related gene (*Lyz1*) in *C. gattii*-infected BMDMs was assessed using quantitative real-time PCR. *C. gattii* infection in BMDMs induced the mRNA expression of several proinflammatory markers, including *Nos2*, *Il6*, and *Tnfa* and treatment with lovastatin effectively increased the mRNA expression of *Nos2* and *Il6* (Fig 4A). Furthermore, the effect of lovastatin on ROS production in *C. gattii*-infected BMDMs was examined using H2DCF-DA to quantify the ROS levels using flow cytometry. We found that lovastatin treatment did not enhance intracellular ROS production in *C. gattii*-infected BMDMs (Fig 4B). Our findings indicated that lowering cholesterol levels using lovastatin enhanced macrophage function during *C. gattii* infection by augmenting the expression of effector molecule Nos2 and inflammatory mediator IL-6 in macrophages.

## Treatment with lovastatin in macrophages enhanced the anti-fungal effect of fluconazole during *C. gattii* infection

Because lovastatin attenuated *C. gattii* infection in macrophages, we evaluated its potential to enhance the antifungal efficacy of fluconazole in *C. gattii*-infected macrophages. BMDMs infected with *C. gattii* were treated with fluconazole alone or in combination with lovastatin, and the ICL, IPR, and extracellular yeast were assessed at 24 hpi. Consistent with our previous data, lovastatin alone significantly reduced ICL and IPR (Fig 5A-B). Compared to the vehicle control, treatment with fluconazole alone or in combination with lovastatin significantly reduced ICL, IPR, and extracellular yeast in *C. gattii*-infected BMDMs (Fig 5A-C). Interestingly, the combination treatment resulted in the greatest reduction in ICL and IPR

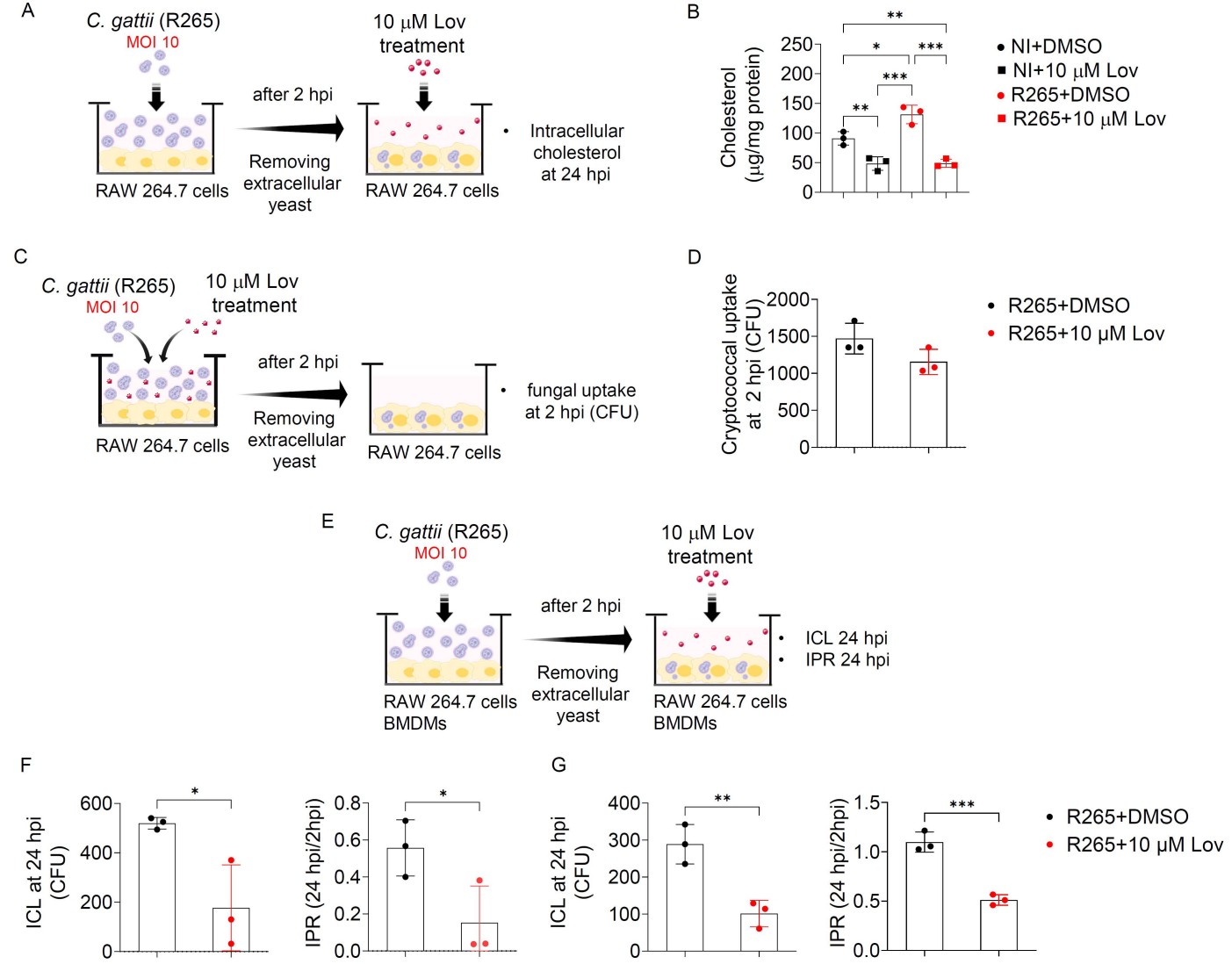

**Fig 3. The effect of lovastatin on cryptococcal uptake and intracellular proliferation in macrophages infected with *C. gattii*.** (A) RAW 264.7 cells were infected with *C. gattii* R265. After 2 hours of infection, non-phagocytosed yeast cells were removed, and macrophages were subsequently treated with either DMSO or 10 μM lovastatin, followed by measuring intracellular cholesterol content at 24 hpi. (B) The level of cholesterol (μg/mg protein) in RAW 264.7 cells from different conditions. (C) RAW 264.7 cells were infected with *C. gattii* R265 and treated with either vehicle control (DMSO) or 10 μM lovastatin for 2 hours. Phagocytosis was then measured using a CFU assay to assess cryptococcal uptake in lovastatin-treated cells compared to the vehicle-treated control. (D) The CFU of *C. gattii* internalized by RAW 264.7 cells at 2 hpi were compared between lovastatin-treated and vehicle control-treated conditions. (E) RAW 264.7 cells or BMDMs were infected with *C. gattii* R265. Two hours post-infection, non-phagocytosed yeast cells were removed, and macrophages were subsequently treated with either DMSO or 10 μM lovastatin, followed by the evaluation of ICL and IPR at 24 hpi using CFU assay. (F-G) The intracellular cryptococcal load (ICL) at 24 hpi and the intracellular proliferation rate (IPR) in (F) RAW 264.7 macrophages and (G) BMDMs. Data for ICL at 24 hpi are presented as CFU, and IPR is expressed as the ratio of CFU at 24 hpi to CFU at 2 hpi. Graphs depict mean±SD of three independent experiments. Significance was determined using one-way ANOVA followed by Turkey post hoc analysis (Fig 3B) and Student's t-test analysis (Fig 3D, 3F-G) (*$p < 0.05$, **$p < 0.01$, ***$p < 0.001$). Experimental conditions: NI+DMSO: Non-infected cells treated with DMSO, NI+10 μM Lov: Non-infected cells treated with 10 μM lovastatin, R265+DMSO: Cells infected with *C. gattii* strain R265 and treated with DMSO, R265+10 μM Lov: Cells infected with *C. gattii* strain R265 and treated with 10 μM lovastatin.

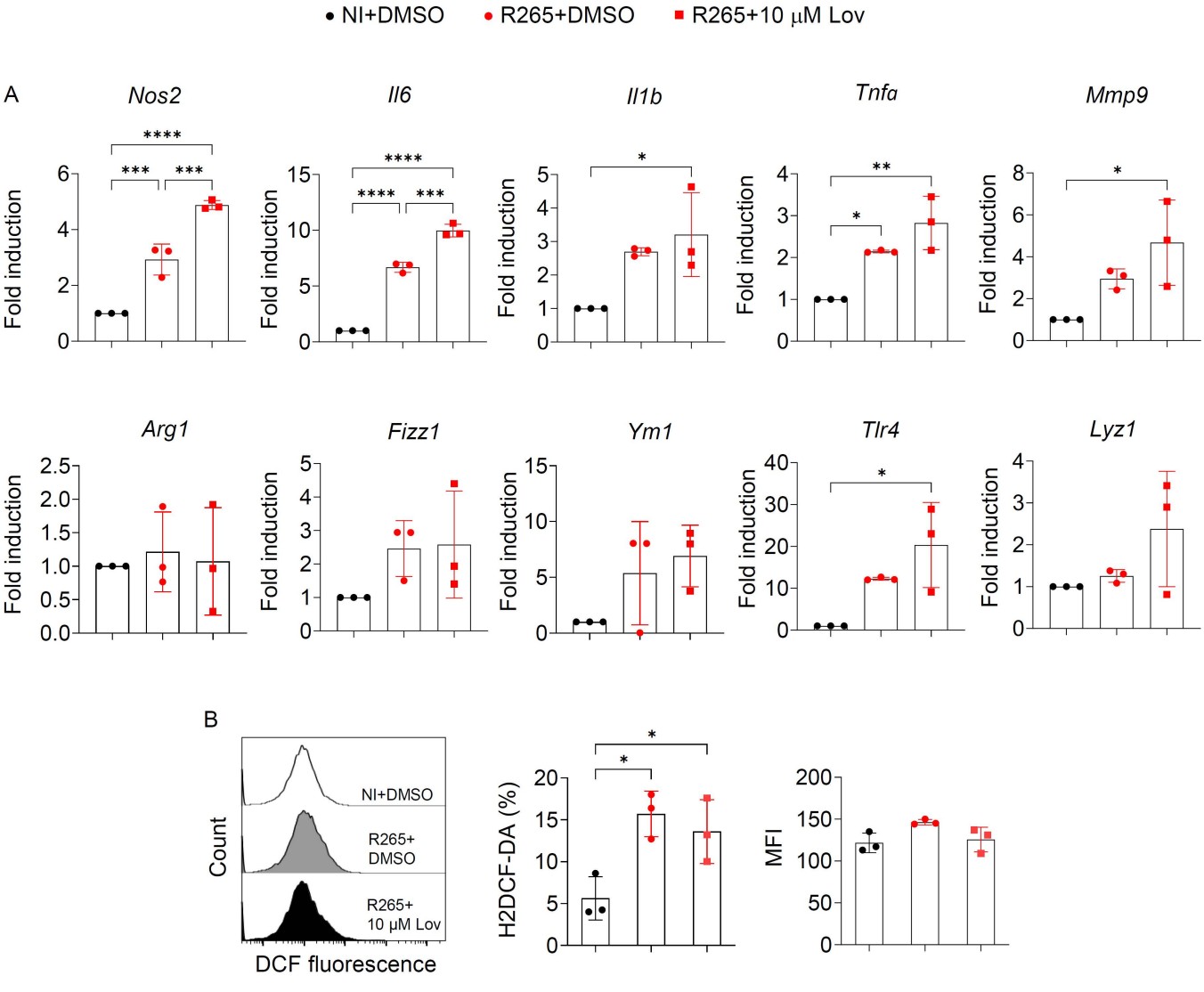

**Fig 4. Lovastatin treatment in macrophages during *C. gattii* R265 infection enhanced the expression of Nos2 and IL-6.** BMDMs were infected with *C. gattii* R265. At 2 hpi, non-phagocytosed yeast cells were removed, and the BMDMs were subsequently treated with either vehicle control (DMSO) or 10 µM lovastatin for 24 hours, followed by mRNA expression analysis. (A) Quantitative real-time PCR analysis of proinflammatory markers (*Nos2*, *Il6*, *Il1b*, *Tnfa*, *Mmp9*, *Tlr4*), anti-inflammatory markers (*Arg1*, *Fizz1*, *Ym1*) and lysosome-related gene (*Lyz1*). Data are presented as fold induction relative to actin (*Actb*) expression, with mRNA levels in non-infected BMDMs treated with DMSO normalized to 1. (B) Representative histogram, percentage, and mean fluorescence intensity (MFI) of ROS production using H2DCF-DA in BMDMs by flow cytometric analysis. Graphs depict mean ± SD of three independent experiments. Significance was determined using one-way ANOVA followed by Turkey post hoc analysis (*$p < 0.05$, **$p < 0.01$, ***$p < 0.001$, ****$p < 0.0001$). Experimental conditions: NI+DMSO: Non-infected cells treated with DMSO, R265 + DMSO: Cells infected with *C. gattii* strain R265 and treated with DMSO, R265 + 10 µM Lov: Cells infected with *C. gattii* strain R265 and treated with 10 µM lovastatin.

compared to fluconazole treatment alone (Fig 5A-B). To assess the potential synergistic effect on inflammatory responses, we measured IL-6 secretion and nitric oxide (NO) activity in BMDM culture supernatants. Consistent with its role in upregulating the gene expression of IL-6 and Nos2 in macrophages, lovastatin enhanced IL-6 secretion and nitric oxide activity in *C. gattii*-infected BMDMs (Fig 5D-E), but did not affect IL-12 or MIP-1α levels (S2 Fig). Interestingly, the combination treatment further increased nitric oxide activity (Fig 5E). In contrast, in *C. neoformans*-infected BMDMs, treatment with

● R265+DMSO      ● R265+10 μM Lov      ● R265+20 μg/mL FLC      ■ R265+20 μg/mL FLC+10 μM Lov

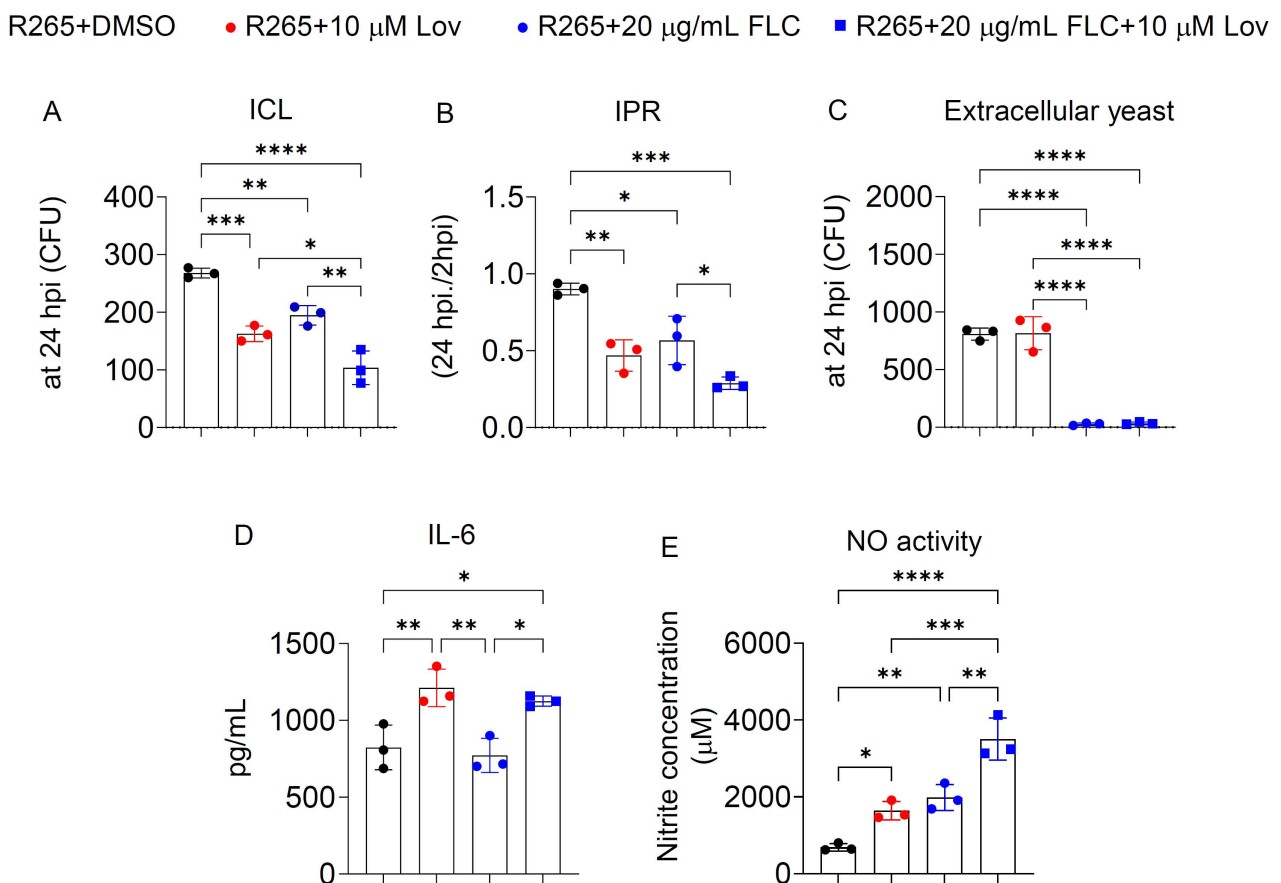

**Fig 5. Treatment with lovastatin potentiated the anti-fungal effect of fluconazole in *C. gattii*-infected BMDMs.** BMDMs were infected with *C. gattii* R265. Two hours post-infection, non-phagocytosed yeast cells were removed, and macrophages were subsequently treated with DMSO, lovastatin (10 μM), fluconazole (20 μg/mL), or combination of fluconazole (20 μg/mL) and lovastatin (10 μM) followed by the evaluation of ICL, IPR, and extracellular yeast at 24 hpi by the CFU assay. (A) The intracellular cryptococcal load (ICL) at 24 hpi in BMDMs. Data are presented as CFU. (B) The intracellular proliferation rate (IPR) in BMDMs. Data are presented as the ratio of CFU at 24 hpi to CFU at 2 hpi. (C) The extracellular yeasts at 24 hpi in BMDMs. Data are presented as CFU. (D) ELISA analysis of IL-6 secretion in culture supernatant of BMDMs treated with indicated treatment conditions. (E) Nitric Oxide (NO) activity was detected in culture supernatant of BMDMs treated with indicated treatment conditions. The y-axis indicated the concentration of nitrite (μM). Graphs depict mean±SD of three independent experiments. Significance was determined using one-way ANOVA followed by Turkey post hoc analysis (*$p<0.05$, **$p<0.01$, ***$p<0.001$, ****$p<0.0001$). Experimental conditions: R265+DMSO: R265-infected cells treated with DMSO, R265+10 μM Lov, R265-infected cells treated with 10 μM lovastatin, R265+20 μg/ml FLC, R265-infected cells treated with 20 μg/ml fluconazole, R265+20 μg/mL FLC+10 μM Lov, R265-infected cells treated with fluconazole (20 μg/mL) and lovastatin (10 μM).

either lovastatin or fluconazole alone reduced ICL and IPR; however, no synergistic effect was observed, and lovastatin alone did not enhance IL-6 or NO activity (S3 Fig). These findings suggest that lovastatin may enhance the anti-fungal effect of fluconazole by promoting macrophage-mediated clearance of *C. gattii* and increasing nitric oxide activity.

### Lovastatin enhances fluconazole-mediated lung fungal clearance by modulating host nitric oxide response in *C. gattii*-infected mice

Our above data demonstrated that lovastatin exerts a synergistic effect with fluconazole in enhancing intracellular clearance of *C. gattii* in macrophages under in vitro conditions. To evaluate its in vivo effects, BALB/c mice were administered lovastatin daily via intraperitoneal injection, either alone or in combination with fluconazole, during *C. gattii* infection

(Fig 6A). At 7 days post-infection, lung fungal burden and inflammatory cell infiltration in bronchoalveolar lavage (BAL) fluid were assessed. Lovastatin treatment significantly reduced lung fungal burden compared to vehicle control (Fig 6B). Consistent with in vitro findings, the combination treatment with lovastatin and fluconazole resulted in the greatest reduction in lung fungal burden (Fig 6B). Moreover, the combination treatment significantly increased total number of BAL cells

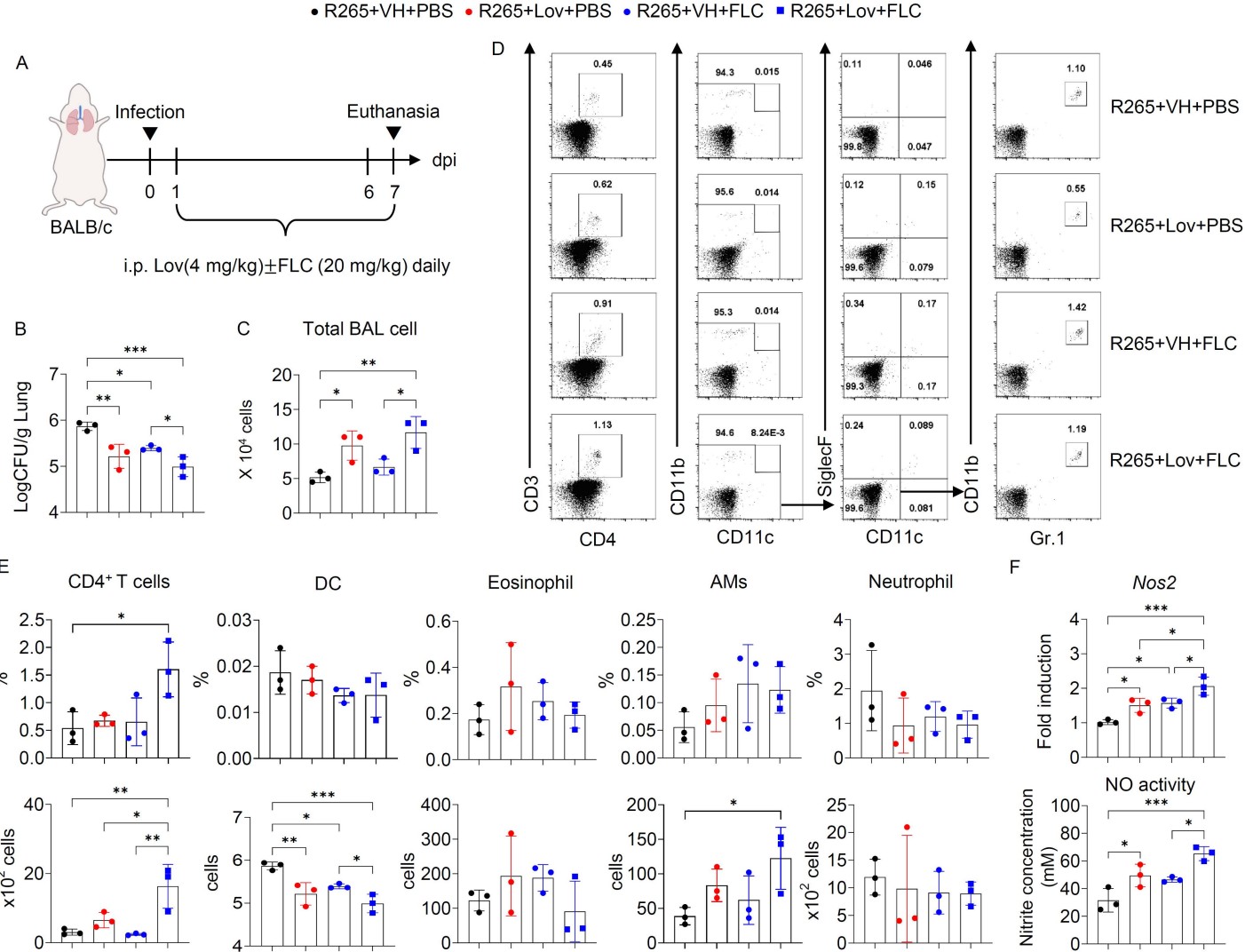

**Fig 6. Lovastatin enhances the antifungal efficacy of fluconazole and promotes host immune responses during *C. gattii* infection.** (A) BALB/c mice were intranasally infected with *C. gattii* R265 and treated daily via intraperitoneal injection with vehicle control (VH), lovastatin (Lov; 4 mg/kg/mouse), fluconazole (FLC; 20 mg/kg/mouse), or combination of both drugs for 6 days. At 7 days post-infection, lungs and BAL samples were collected and analyzed for fungal burden (CFU), BAL cell infiltration, Nos2 gene expression, and NO activity. (B) Lung fungal burden was quantified. (C) Total BAL cell numbers were enumerated. (D) Representative flow cytometry gating strategies were used to identify inflammatory cell populations, including Th cells (CD3⁺ CD4⁺), dendritic cells (DC; CD11b⁺ CD11c⁺), eosinophils (CD11c⁻ Siglec-F⁺), alveolar macrophages (CD11c⁺ Siglec-F⁺), and neutrophils (CD11b⁺ Gr-1$^{hi}$). (E) Representative data showing the percentages and absolute numbers of inflammatory cells in BAL fluid. (F) Quantitative real-time PCR analysis of *Nos2* in the lung tissue, presented as fold induction over actin (*Actb*) expression, with the mRNA levels in the vehicle control group set as 1 (upper), and Nitric Oxide (NO) activity was detected in BAL fluid (lower). The y-axis indicated the concentration of nitrite (mM). Graphs depict individual mice with the mean ± SD of a representative from two independent experiments (n = 3 mice per group). Significance was determined using one-way ANOVA followed by Tukey's multiple comparisons test (*$p < 0.05$, **$p < 0.01$, ***$p < 0.001$).

compared to fluconazole alone (Fig 6C). Interestingly, it also enhanced both the percentage and absolute number of CD4+ T cells (Fig 6D-E). Given that the combination treatment enhanced nitric oxide activity in *C. gattii*-infected macrophages in vitro, we also evaluated lung *Nos2* gene expression and nitric oxide activity in BAL fluid. The combination treatment exhibited the greatest increase in lung *Nos2* gene expression and activity (Fig 6F). These findings suggest that exogenous lovastatin enhances lung fungal clearance and potentiates the antifungal effect of fluconazole by augmenting host nitric oxide activity.

## Discussion

Alveolar macrophages are key players in early defense against cryptococcal infection; however, the mechanisms by which different *Cryptococcus* species, especially *C. gattii* infection, modulate alveolar macrophage function remain largely unknown. Our study revealed distinct transcriptional profiles in alveolar macrophages following *C. gattii* and *C. neoformans* infection, with *C. gattii* inducing genes related to sterol biosynthesis pathway and *C. neoformans* activating genes related to interferon and cytokine signaling pathways. Inhibition of sterol biosynthesis by lovastatin reduced both intracellular cryptococcal load and proliferation rate in *C. gattii*-infected RAW 264.7 and BMDMs, while concurrently enhancing the expression of *Nos2* and *Il6*. Lovastatin also synergized with fluconazole to suppress intracellular fungal growth and enhance nitric oxide activity in macrophages infected with *C. gattii*. In vivo, the combined treatment improved pulmonary fungal clearance and augmented host immune response by increasing lung CD4+ T cell infiltration and nitric oxide activity, highlighting sterol metabolism as a potential therapeutic target.

Previous study has shown that transcriptomic profiles of macrophages vary depending on cell type, activation state, and experimental conditions [21]. However, the in vivo dynamics of alveolar macrophages in response to different pathogenic *Cryptococcus* spp. remain poorly understood. In this study, we compared gene expression in alveolar macrophages from mice infected with *C. gattii* R265 and *C. neoformans* H99. *C. gattii* infection led to significant upregulation of sterol biosynthesis-related genes, including *Fdps*, *Cyp51*, *Sqle*, *Hsd17b7*, and *Lss*. Previous study in RAW 264.7 macrophages has shown that inhibition of sterol biosynthesis impairs TLR signaling and cytokine production in macrophages under lipid-depleted conditions [41], highlighting the essential role of sterol biosynthesis in macrophage functionality related to inflammation. Besides the sterol biosynthesis pathway, we found an increase in genes associated with the synthesis of organic hydroxy compound, small molecule biosynthetic process, and leukocyte chemotaxis. Among these upregulated genes, prior research demonstrated that *Nr4a2* has been linked to M2 polarization and protection in sepsis models [42], while *Ccl17* and *Csf1* are implicated in the recruitment of Th2 cells [43,44]. Further investigation is needed to determine how these pathways influence macrophage response to *C. gattii* infection. Conversely, majority of the downregulated genes were linked to the regulation of plasma lipoproteins, cholesterol efflux, and lipid particles. Previous studies indicated that some intracellular pathogens accumulated intracellular cholesterol as an energy source to support their survival [45–47]. For example, *Mycobacterium tuberculosis* can utilize cholesterol within host macrophages to persist in IFNγ-activated macrophages [45]. BMDMs infected with *Salmonella* Typhimurium exhibited downregulation of ABCA expression and heightened cholesterol accumulation, facilitating intracellular bacterial survival [46]. Additionally, transcriptomic analysis of *C. gattii* biofilm revealed a metabolic shift favoring alternative carbon sources such as fatty acids, amino acids, acetate, or lactate during biofilm formation, alongside with the upregulation of sterol biosynthesis genes essential for fungal survival [47]. Together, our findings highlight that *C. gattii* infection uniquely alters alveolar macrophages, primarily through modulation of sterol metabolism.

Unlike AMs from *C. gattii*-infected mice, those infected with *C. neoformans* exhibited an upregulation of genes mostly associated with host immune responses, particularly interferon signaling. The role of type I IFN in *C. neoformans* infection remains controversial. While type I IFN signaling has been linked to impaired fungal clearance [48], it also appears essential for Th17 polarization and early immune control [49]. Additional research is required to elucidate the precise function of type I interferon in alveolar macrophages during *C. neoformans* infection. In addition, *C. neoformans* infection

resulted in downregulation of genes involved in oxygen binding and hemoglobin synthesis (*Hbb-bs*, *Hba-a1*, and *Hba-a2*) in alveolar macrophages. Prior research suggested that the reduced expression of hemoglobin-related genes in alveolar macrophages of *C. neoformans*-infected mice may be associated with the oxygen-dependent killing mechanisms of these cells [25]. However, further investigation is needed to clarify the specific role of these genes during *C. neoformans* infection. Altogether, our findings emphasize that *C. neoformans* modulates alveolar macrophages distinctively compared to *C. gattii* infection, mainly by affecting genes related to the immune response regulation.

Given the observed upregulation of sterol biosynthesis genes in alveolar macrophages during *C. gattii* infection, we investigated whether cholesterol-lowering drugs could modulate intracellular fungal proliferation in macrophages. Statins, widely prescribed for hypercholesterolemia and dyslipidemia [50], inhibit 3-hydroxy-3-methylglutaryl coenzyme A (HMG-CoA) reductase, thereby blocking the conversion of HMG-CoA to mevalonate and suppressing cholesterol biosynthesis [51,52]. Our findings demonstrate that treatment of macrophages with lovastatin during *C. gattii* infection significantly reduced intracellular fungal proliferation, accompanied by increased IL-6 secretion and Nos2 expression. These results suggest that enhanced sterol synthesis triggered by *C. gattii* infection may facilitate fungal survival by dampening macrophage inflammatory responses and effector functions. Interestingly, previous studies have reported that statin use is associated with a reduced incidence of influenza and fungal infections [53], as well as reduced mortality of patients hospitalized with infection-associated pneumonia [54]. Although lovastatin did not exhibit direct antifungal activity against *C. gattii* at the concentration used in our study, this may be due to the lower dosage, as previous reports have shown antifungal effects of lovastatin against *M. racemosus* [36] and *C. albicans* [37,38] at higher concentration. Thus, at the tested concentration, lovastatin likely exerts its effects primarily through modulation of macrophage function rather than direct fungal inhibition.

To further investigate the potential therapeutic synergy, we examined the combined effect of lovastatin and the antifungal agent fluconazole. Combined lovastatin and fluconazole treatment significantly reduced intracellular *C. gattii* growth and increased nitric oxide production in macrophages. In vivo, lovastatin enhanced fluconazole efficacy by improving fungal clearance, inflammatory cell infiltration, and nitric oxide activity in the lungs. A related study reported that combined therapy of atorvastatin and fluconazole had synergistic effects against *C. gattii* infection and improved survival of infected mice, supporting the potential use of statins as adjunctive therapy [28]. Further studies are needed to evaluate the effects of different statins and to confirm the role of sterol metabolism in modulating host immune response against *C. gattii* infection, especially in vivo. Although lovastatin also reduced intracellular fungal growth in *C. neoformans*-infected macrophages, no synergistic effect with fluconazole was observed. This may reflect a more robust activation of the sterol biosynthesis pathway during host interaction with *C. gattii*, potentially due to differences in stress response, membrane remodeling, and metabolic adaptation [47,55,56]. Future research should focus on comparative metabolomic profiling under host-mimicking conditions to elucidate the regulatory mechanisms driving sterol pathway activation in *C. gattii*, and to determine its role in pathogenesis, immune modulation, and antifungal resistance.

## Conclusion

In summary (Fig 7), this study delineated the unique transcriptional patterns of alveolar macrophages in response to *C. neoformans* and *C. gattii* infections, expanding the understanding of how sterol biosynthesis pathways, induced by *C. gattii* infection, may modulate macrophage function during infection. Modulation of macrophage cholesterol via lovastatin attenuated the intracellular fungal proliferation and enhanced inflammatory cytokine IL-6 secretion and nitric oxide activity. Treatment of lovastatin enhanced the antifungal efficacy of fluconazole and augmented nitric oxide activity in macrophages. These data suggest that suppressing the sterol synthesis triggered by *C. gattii* infection, in conjunction with antifungal drugs, may enhance the anti-fungal effect by augmenting macrophage effector function, hence offering intriguing alternative strategy for more effective therapy.

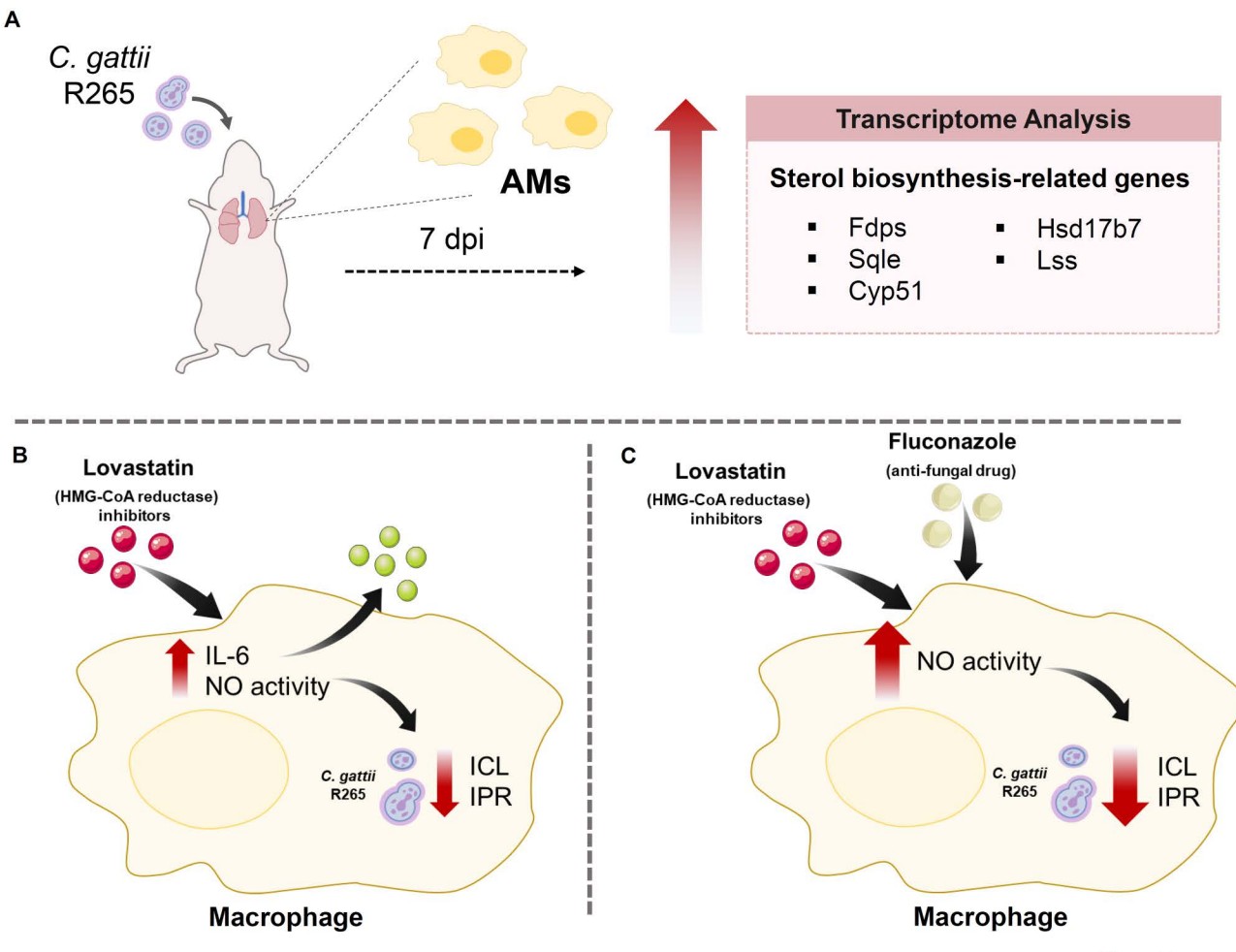

Figure 7

**Fig 7. A proposed role of sterol metabolism in macrophage response to *C. gattii* infection.** (A) Transcriptomic analysis reveals the upregulation of genes related with sterol biosynthesis in AMs during the early stage of *C. gattii* infection. (B) Lovastatin treatment inhibits the intracellular cryptococcal proliferation and augments the secretion of IL-6 and nitric oxide activity in macrophages. (C) The combined treatment of lovastatin and fluconazole exhibits a synergistic effect in promoting anti-fungal activity, coupled with enhancing nitric oxide production in macrophages.

## Supporting information

**S1 Table. Data quality summary of purified alveolar macrophage samples from PBS-treated, *C. gattii*-infected and *C. neoformans*-infected mice at 7 days after infection.**
(DOCX)

**S2 Table. Primers used for quantitative real-time PCR analysis.**
(DOCX)

**S3 Table. List of the top 20 significantly upregulated and downregulated DEGs comparing lung AMs from mice infected with *C. neoformans* at 7 dpi to those treated with PBS.**
(DOCX)

**S4 Table. List of the top 20 significantly upregulated and 3 significantly downregulated DEGs comparing lung AMs from mice infected with *C. gattii* at 7 dpi to those treated with PBS.**
(DOCX)

**S5 Table. List of the significant GO enrichment analyses of upregulated DEGs comparing lung AMs from mice infected with *C. gattii* to those treated with PBS.**
(DOCX)

**S6 Table. List of the most significant GO enrichment analyses of downregulated DEGs comparing lung AMs from mice infected with *C. gattii* to those treated with PBS.**
(DOCX)

**S1 Fig. Direct effect of lovastatin treatment on the in vitro growth of *C. gattii* and *C. neoformans*.**
(DOCX)

**S2 Fig. IL-12 and MIP-1α secretion in *C. gattii*-infected BMDM culture supernatants.**
(DOCX)

**S3 Fig. Effect of lovastatin and its combination with fluconazole on *C. neoformans*-infected BMDMs.**
(DOCX)

**S1 Data. Raw data.**
(XLSX)

**S1 Checklist. Plos one endpoints human Checklist.**
(DOCX)

## Acknowledgments

We thank the Faculty of Allied Health Sciences and Center of Scientific Equipment for Advanced Research for their support.

## Author contributions

**Conceptualization:** Pornpimon Angkasekwinai.

**Data curation:** Siranart Jeerawattanawart, Yongliang Zhang, Pornpimon Angkasekwinai.

**Formal analysis:** Siranart Jeerawattanawart, Adithap Hansakon, Rudi Alberts, Pornpimon Angkasekwinai.

**Funding acquisition:** Pornpimon Angkasekwinai.

**Investigation:** Siranart Jeerawattanawart, Adithap Hansakon, Rudi Alberts, Pornpimon Angkasekwinai.

**Methodology:** Siranart Jeerawattanawart, Adithap Hansakon, Pornpimon Angkasekwinai.

**Project administration:** Pornpimon Angkasekwinai.

**Resources:** Yongliang Zhang.

**Supervision:** Pornpimon Angkasekwinai.

**Validation:** Siranart Jeerawattanawart, Pornpimon Angkasekwinai.

**Visualization:** Siranart Jeerawattanawart, Pornpimon Angkasekwinai.

**Writing – original draft:** Siranart Jeerawattanawart, Pornpimon Angkasekwinai.

**Writing – review & editing:** Siranart Jeerawattanawart, Pornpimon Angkasekwinai.

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
