## [Decision Letter · Decision Letter 0]

26 Jun 2025

PONE-D-25-29534Transcriptomic profiling of lung alveolar macrophages reveals distinct contribution of sterol metabolism in macrophage response to Cryptococcus gattii infectionPLOS ONE

Dear Dr. Angkasekwinai, Thank you for submitting your manuscript to PLOS ONE. After careful consideration, we feel that it has merit but requires some additional information/changes. Therefore, we invite you to submit a revised version of the manuscript that addresses the points raised during the review process.

All the reviewers agree that the data presented on the differential transcriptional expression signatures of alveolar macrophages in mice infected with *Cryptococcus neoformans* and *C. gattii* are novel and potentially translatable. The reviewers also believe that the experiments are well-designed and the results are clearly presented and has significant impact to the field.

However, each reviewer has raised specific concerns that need to be addressed. Please respond to these comments in a timely manner.In addition to the reviewers' feedback, I have one minor comment regarding Figure 1. As currently shown, it appears that the mice were infected with both *C. neoformans* and *C. gattii* . To improve clarity, consider adding the word “or” between the two yeast strains in the figure to emphasize that the infection was with one strain or the other—not a mixture of both.

We look forward to receiving your revised manuscript.

Kind regards,

Rajendra Upadhya

Academic Editor

PLOS ONE

Journal Requirements:

“This study was supported by the Thailand Science Research and Innovation (TSRI) Fundamental Fund fiscal year 2024, Thammasat University and the Thammasat University Research Unit in Molecular Pathogenesis and Immunology of Infectious Diseases.”

“This study was supported by the Thailand Science Research and Innovation (TSRI) Fundamental Fund fiscal year 2024, Thammasat University and the Thammasat University Research Unit in Molecular Pathogenesis and Immunology of Infectious Diseases. We thank the Faculty of Allied Health Sciences and Center of Scientific Equipment for Advanced Research, for their support.”

“This study was supported by the Thailand Science Research and Innovation (TSRI) Fundamental Fund fiscal year 2024, Thammasat University and the Thammasat University Research Unit in Molecular Pathogenesis and Immunology of Infectious Diseases.”

Reviewers' comments:

Reviewer's Responses to Questions

**Comments to the Author**

1. Is the manuscript technically sound, and do the data support the conclusions?

Reviewer #1: Yes

Reviewer #2: Yes

Reviewer #3: Yes

2. Has the statistical analysis been performed appropriately and rigorously? 

Reviewer #1: Yes

Reviewer #2: Yes

Reviewer #3: Yes

3. Have the authors made all data underlying the findings in their manuscript fully available?

Reviewer #1: No

Reviewer #2: Yes

Reviewer #3: Yes

4. Is the manuscript presented in an intelligible fashion and written in standard English?

Reviewer #1: Yes

Reviewer #2: Yes

Reviewer #3: Yes

5. Review Comments to the Author

Reviewer #1: The authors have conducted an excellent study, demonstrating that C. gattii infection induces the sterol biosynthesis pathway in alveolar macrophages (AMs), whereas C. neoformans primarily activates immune signaling, highlighting distinct host response mechanisms between the two fungi. Additionally, inhibiting sterol metabolism (e.g., with lovastatin) enhances the antifungal capacity of macrophages and synergizes with fluconazole, proposing a novel therapeutic strategy for cryptococcal infections.

However, I suggest several modifications to strengthen the persuasiveness of the findings:

Major Revisions

1.In the experiments examining the inflammatory response of macrophages during cryptococcal infection with lovastatin, why was only IL-6 measured? Measuring other pro-inflammatory cytokines (e.g., TNF-α, IL-1β) would render the conclusions of this section more convincing.

2.As noted by the authors, statins have exhibited antifungal effects against other fungi. Their synergistic effects with antifungal drugs in C. neoformans treatment have also been reported (PMID: 32321104). It is recommended to supplement experiments evaluating the effect of statins on C. neoformans and their synergism with antifungal agents. A detailed discussion on the similarities and differences in mechanisms between C. gattii and C. neoformans is warranted.

3.In the Discussion, the authors found that the sterol metabolism pathway is significantly activated in C. gattii infection but not in C. neoformans. What might be the underlying reasons for this discrepancy? This section should be expanded to explore potential mechanisms, such as differences in fungal virulence factors or host receptor recognition.

4.The conclusion that statins enhance macrophage antifungal activity and synergize with fluconazole is promising for novel therapeutic strategies in cryptococcal infections. However, current evidence is limited to in vitro experiments. Including in vivo data (e.g., mouse infection models) would strengthen the translational relevance and clinical confidence in this approach.

5.All raw data for statistical analyses should be provided for review. This includes detailed numerical values, sample sizes, and statistical parameters to validate the results.

These revisions will enhance the rigor and translational impact of the study. Please address each point systematically and provide clarifications where necessary.

Minor Revisions

Line 310, change "differentially expressed significant genes" to "significantly differentially expressed genes".

Reviewer #2: This study submitted by Siranart Jeerawattanawartand collaborators, elucidated the unique transcriptional profiles of alveolar macrophages in response to infections with Cryptococcus neoformans and C. gattii, expanding our understanding of the mechanisms by which C. gattii-induced sterol biosynthesis pathways modulate macrophage function during infection. Modulation of macrophage cholesterol metabolism using lovastatin led to a reduction in the intracellular proliferation of the fungus, accompanied by a significant increase in IL-6 production and nitric oxide activity. Furthermore, lovastatin treatment enhanced the antifungal efficacy of fluconazole, further amplifying the nitric oxide-mediated inflammatory response. These findings indicate that inhibition of sterol biosynthesis triggered by C. gattii infection, in combination with antifungal agents, may enhance therapeutic efficacy by boosting macrophage effector functions. This strategy thus represents a promising alternative approach to optimize antifungal therapy against Cryptococcus infections.

Transcriptomic analysis revealed an early upregulation of genes involved in sterol biosynthesis in macrophages during Cryptococcus gattii infection. Treatment with lovastatin reduced the intracellular proliferation of Cryptococcus and promoted both IL-6 secretion and nitric oxide production by macrophages. Moreover, the combination of lovastatin and fluconazole exhibited a synergistic antifungal effect, accompanied by increased nitric oxide production.

The manuscript is very well written, and the results are clear, thoroughly addressing the scientific objectives of the study. The data are highly interesting and hold significant relevance for the field. However, I have two minor reservations:

a) A manuscript focusing on Cryptococcus would benefit from an introduction that highlights a key characteristic of the fungus—the polysaccharide capsule. Below are two relevant references for inclusion:

a) Decote-Ricardo D, LaRocque-de-Freitas IF, Rocha JDB, Nascimento DO, Nunes MP, Morrot A, Freire-de-Lima L, Previato JO, Mendonça-Previato L, Freire-de-Lima CG. Immunomodulatory Role of Capsular Polysaccharides Constituents of Cryptococcus neoformans. Front Med. 2019;6:129. doi:10.3389/fmed.2019.00129.

b) Fonseca FL, Reis FCG, Sena BAG, Jozefowicz LJ, Kmetzsch L, Rodrigues ML. The Overlooked Glycan Components of the Cryptococcus Capsule. Curr Top Microbiol Immunol. 2019;422:31-43. doi:10.1007/82_2018_140.

Another suggestion would be a recently published manuscript that shows that infection by Cryptococcus gattii more strongly impairs the visual capacity of infected mice; this citation could be included among references 18–20, cited in lines 69–70 (see reference below).

a) Lack of TLR9 exacerbates ocular impairment and visual loss during systemic Cryptococcus gattii infection. da Silva-Junior EB, de Araujo VG, Araujo-Silva CA, Covre LP, Guimarães de Oliveira JC, Diniz-Lima I, Freire-de-Lima L, Morrot A, de Brito-Gitirana L, Previato JO, Mendonça-Previato L, Vommaro RC, Petrs-Silva H, Decote-Ricardo D, Guedes HLM, Freire-de-Lima CG.J Infect Dis. 2025 Jan 15:jiaf034. doi: 10.1093/infdis/jiaf034.

Reviewer #3: C.neoformans and C.gattii which have been classified as priority fungal pathogens cause worldwide health-threatening impact. The manuscript delineated the unique transcriptional patterns of alveolar macrophages in response to C.neoformans and C.gattii infections. Results reveal the distinct response of alveolar macrophages to C.gattii infection and the relation of macrophage sterol metabolism in modulating C.gattii infection. This study provides potential strategy for the treatment of cryptococcal infections. However, several areas could benefit from improvements. Blow are comments and suggestions.

Comments:

1. Line 90-94: These descriptions are the results of the study, please remove them. The concise methods and overarching goal of this research should be detailed in the introduction.

2. Place the Fig.1 to Fig. 5 sections (Line 325-337, Line 370-385, Line 415-432, Line 450-461, and Line485-498) in the Figure legends for description.

3. The descriptions of the results were too redundant , some parts of the results could be removed to the methods. For example, line 469 to line 470.

4. Please definite the units of longitudinal coordinates in Figure 2C, Figure 3C, Figure 3D, Figure 4 and Figure 5.

6. PLOS authors have the option to publish the peer review history of their article (what does this mean? ). If published, this will include your full peer review and any attached files.

**Do you want your identity to be public for this peer review?** For information about this choice, including consent withdrawal, please see our Privacy Policy .

Reviewer #1: No

Reviewer #2: No

Reviewer #3: **Yes: ** Yan Cheng (Department of Clinical Laboratory, Bethune International Peace Hospital, Shijiazhuang 050081, China)

---

## [Author Response · Author response to Decision Letter 1]

24 Aug 2025

Reviewer 1:

The authors have conducted an excellent study, demonstrating that C. gattii infection induces the sterol biosynthesis pathway in alveolar macrophages (AMs), whereas C. neoformans primarily activates immune signaling, highlighting distinct host response mechanisms between the two fungi. Additionally, inhibiting sterol metabolism (e.g., with lovastatin) enhances the antifungal capacity of macrophages and synergizes with fluconazole, proposing a novel therapeutic strategy for cryptococcal infections.

However, I suggest several modifications to strengthen the persuasiveness of the findings:

Major Revisions

1. In the experiments examining the inflammatory response of macrophages during cryptococcal infection with lovastatin, why was only IL-6 measured? Measuring other pro-inflammatory cytokines (e.g., TNF-α, IL-1β) would render the conclusions of this section more convincing.

Response: We thank reviewer’s constructive critics. In this study, we examined the gene expression of cytokines involved in the inflammatory response of macrophages during cryptococcal infection with lovastatin treatment using quantitative real-time PCR. Our initial results showed that lovastatin did not alter the mRNA expression of either Tnfa or Il1b in BMDMs. Therefore, we further performed additional experiments to assess the secretion of other cytokines and chemokines related to inflammation response, including IL-12 and MIP-1α in the culture supernatant of BMDMs from different treatment groups using the Bio-Plex multiplex assay. Our results revealed that lovastatin treatment did not alter the secretion of MIP-1� and IL-12 in R265-infected BMDMs, suggesting that lovastatin may specifically influence IL-6 regulation in R265-infected BMDMs. These additional findings are presented in Supplementary Fig. S2 and included in lines 527-528 of the revised manuscript.

2. As noted by the authors, statins have exhibited antifungal effects against other fungi. Their synergistic effects with antifungal drugs in C. neoformans treatment have also been reported (PMID: 32321104). It is recommended to supplement experiments evaluating the effect of statins on C. neoformans and their synergism with antifungal agents. A detailed discussion on the similarities and differences in mechanisms between C. gattii and C. neoformans is warranted.

Response: We appreciate the reviewer’s comment. We performed an additional investigation into the potential synergistic effect of lovastatin and fluconazole treatment in C. neoformans-infected BMDMs. In C. neoformans-infected BMDMs, treatment with either lovastatin or fluconazole alone reduced ICL and IPR; however, no synergistic effect was observed, and lovastatin alone did not enhance IL-6 or NO activity, suggesting that lovastatin’s immunomodulatory effects on macrophage may be less pronounced in C. neoformans infection. We have included additional data in Supplementary Fig. S3 and in lines 529-531 and also discussed the differences in mechanisms between C. gattii and C. neoformans in lines 670-674 in the revised manuscript.

3. In the Discussion, the authors found that the sterol metabolism pathway is significantly activated in C. gattii infection but not in C. neoformans. What might be the underlying reasons for this discrepancy? This section should be expanded to explore potential mechanisms, such as differences in fungal virulence factors or host receptor recognition.

Response: We appreciate the reviewer’s comment. A more robust activation of the sterol biosynthesis pathway during host interaction with C. gattii than in C. neoformans, likely due to species-specific differences in stress response, membrane remodeling, and metabolic adaptation [47, 55, 56]. Future research should focus on comparative metabolomic profiling under host-mimicking conditions to elucidate the regulatory mechanisms driving sterol pathway activation in C. gattii, and to determine its role in pathogenesis and antifungal resistance. We have included the discussion in lines 668-676 in the revised manuscript.

4. The conclusion that statins enhance macrophage antifungal activity and synergize with fluconazole is promising for novel therapeutic strategies in cryptococcal infections. However, current evidence is limited to in vitro experiments. Including in vivo data (e.g., mouse infection models) would strengthen the translational relevance and clinical confidence in this approach.

Response: We appreciate the reviewer’s comment. We conducted additional analyses to evaluate the effect of lovastatin in a mouse model of pulmonary cryptococcosis caused by C. gattii infection. To evaluate its in vivo effects, BALB/c mice were administered lovastatin daily via intraperitoneal injection, either alone or in combination with fluconazole, during C. gattii infection (Fig. 6A). At 7 days post-infection, lung fungal burden and inflammatory cell infiltration in bronchoalveolar lavage (BAL) fluid were assessed. Lovastatin treatment significantly reduced lung fungal burden compared to vehicle control (Fig. 6B). Consistent with in vitro findings, the combination treatment with lovastatin and fluconazole resulted in the greatest reduction in lung fungal burden (Fig. 6B). Moreover, the combination treatment significantly increased total number of BAL cells compared to fluconazole alone (Fig. 6C). Interestingly, it also enhanced both the percentage and absolute number of CD4+ T cells (Fig. 6D-E). Given that the combination treatment enhanced nitric oxide activity in C. gattii-infected macrophages in vitro, we also evaluated lung Nos2 gene expression and nitric oxide activity in BAL fluid. The combination treatment exhibited the greatest increase in lung Nos2 gene expression and activity (Fig. 6F). These findings suggest that exogenous lovastatin enhances lung fungal clearance and potentiates the antifungal effect of fluconazole by augmenting host nitric oxide activity. We have included additional data in Fig. 6 and lines 553-570 in the revised manuscript.

5. All raw data for statistical analyses should be provided for review. This includes detailed numerical values, sample sizes, and statistical parameters to validate the results. These revisions will enhance the rigor and translational impact of the study. Please address each point systematically and provide clarifications where necessary.

Response: We appreciate the reviewer’s comment. In this revised manuscript, we have included raw data as suggested.

Minor Revisions

Line 310, change "differentially expressed significant genes" to "significantly differentially expressed genes".

Response: We appreciate the reviewer’s comment. In the revised manuscript, we have updated the phrase to 'significantly differentially expressed genes' as recommended.

Reviewer #2:

This study submitted by Siranart Jeerawattanawart and collaborators, elucidated the unique transcriptional profiles of alveolar macrophages in response to infections with Cryptococcus neoformans and C. gattii, expanding our understanding of the mechanisms by which C. gattii-induced sterol biosynthesis pathways modulate macrophage function during infection. Modulation of macrophage cholesterol metabolism using lovastatin led to a reduction in the intracellular proliferation of the fungus, accompanied by a significant increase in IL-6 production and nitric oxide activity. Furthermore, lovastatin treatment enhanced the antifungal efficacy of fluconazole, further amplifying the nitric oxide-mediated inflammatory response. These findings indicate that inhibition of sterol biosynthesis triggered by C. gattii infection, in combination with antifungal agents, may enhance therapeutic efficacy by boosting macrophage effector functions. This strategy thus represents a promising alternative approach to optimize antifungal therapy against Cryptococcus infections.

Transcriptomic analysis revealed an early upregulation of genes involved in sterol biosynthesis in macrophages during Cryptococcus gattii infection. Treatment with lovastatin reduced the intracellular proliferation of Cryptococcus and promoted both IL-6 secretion and nitric oxide production by macrophages. Moreover, the combination of lovastatin and fluconazole exhibited a synergistic antifungal effect, accompanied by increased nitric oxide production.

The manuscript is very well written, and the results are clear, thoroughly addressing the scientific objectives of the study. The data are highly interesting and hold significant relevance for the field. However, I have two minor reservations:

a) A manuscript focusing on Cryptococcus would benefit from an introduction that highlights a key characteristic of the fungus—the polysaccharide capsule. Below are two relevant references for inclusion:

a) Decote-Ricardo D, LaRocque-de-Freitas IF, Rocha JDB, Nascimento DO, Nunes MP, Morrot A, Freire-de-Lima L, Previato JO, Mendonça-Previato L, Freire-de-Lima CG. Immunomodulatory Role of Capsular Polysaccharides Constituents of Cryptococcus neoformans. Front Med. 2019;6:129. doi:10.3389/fmed.2019.00129.

b) Fonseca FL, Reis FCG, Sena BAG, Jozefowicz LJ, Kmetzsch L, Rodrigues ML. The Overlooked Glycan Components of the Cryptococcus Capsule. Curr Top Microbiol Immunol. 2019;422:31-43.doi:10.1007/82_2018_140.

Response: We appreciate the reviewer’s insightful comment. We have expanded the introduction to highlight a key characteristic of Cryptococcus spp. Specifically, we added the following statement and references in lines 54–57 of the revised manuscript: “A hallmark of both species is the presence of a polysaccharide capsule composed of glucuronoxylomannan (GXM), galactoxylomannan (GalXM), and glucuronoxylomanogalactan (GXMGal). These capsule components are critical virulence factors that facilitate immune evasion, inhibit phagocytosis, and modulate host immune responses [5, 6].”

Another suggestion would be a recently published manuscript that shows that infection by Cryptococcus gattii more strongly impairs the visual capacity of infected mice; this citation could be included among references 18–20, cited in lines 69–70 (see reference below).a) Lack of TLR9 exacerbates ocular impairment and visual loss during systemic Cryptococcus gattii infection. da Silva-Junior EB, de Araujo VG, Araujo-Silva CA, Covre LP, Guimarães de Oliveira JC, Diniz-Lima I, Freire-de-Lima L, Morrot A, de Brito-Gitirana L, Previato JO, Mendonça-Previato L, Vommaro RC, Petrs-Silva H, Decote-Ricardo D, Guedes HLM, Freire-de-Lima CG.J Infect Dis. 2025 Jan 15:jiaf034. doi: 10.1093/infdis/jiaf034.

Response: In the revised manuscript, we have added further details in the introduction to demonstrate the distinct roles of macrophage subsets in response to different Cryptococcus species. Specifically, in lines 77–79, we included the following statement: “Recent studies have underscored the role of TLR9 expressed in brain microglial cells during C. gattii infection. In TLR9-deficient mice, the absence of this receptor may facilitate the entry of yeast cells into the brain [23, 24].”

Reviewer #3:

C. neoformans and C. gattii, which have been classified as priority fungal pathogens cause worldwide health-threatening impact. The manuscript delineated the unique transcriptional patterns of alveolar macrophages in response to C. neoformans and C. gattii infections. Results reveal the distinct response of alveolar macrophages to C. gattii infection and the relation of macrophage sterol metabolism in modulating C. gattii infection. This study provides potential strategy for the treatment of cryptococcal infections.

However, several areas could benefit from improvements. Below are comments and suggestions.

Comments:

1. Line 90-94: These descriptions are the results of the study, please remove them. The concise methods and overarching goal of this research should be detailed in the introduction.

Response: We appreciate the reviewer’s thoughtful suggestion. We have revised the paragraph in the introduction of this revised manuscript (lines 96-105) as follows: “In this study, we aimed to investigate the transcriptional profiles of alveolar macrophages isolated from mice intranasally infected with C. gattii R265 using RNA sequencing, with the goal of identifying key molecular pathways involved in the host immune response. Based on the observed upregulation of sterol biosynthesis-related genes in alveolar macrophages following C. gattii infection, we further sought to examine the effect of sterol biosynthesis inhibition through lovastatin treatment, both alone and in combination with the anti-fungal drug fluconazole, on macrophage responses in vitro. Additionally, we evaluated the effect of these treatments in a mouse model of pulmonary cryptococcosis. These investigations may advance our understanding of host-pathogen interactions and support the development of potential therapeutic strategies for C. gattii infections.”

2. Place the Fig.1 to Fig. 5 sections (Line 325-337, Line 370-385, Line 415-432, Line 450-461, and Line485-498) in the Figure legends for description.

Response: We appreciate suggestions from the reviewer. In accordance with the PLOS ONE journal submission guidelines, we have revised the figure captions to ensure that each caption appears directly after the paragraph in which the corresponding figure is first cited. Tables have been removed from captions, and figure titles are now presented in bold type. We hope this format remains acceptable and appreciate the reviewer’s understanding.

3. The descriptions of the results were too redundant, some parts of the results could be removed to the methods. For example, line 469 to line 470.

Response: We appreciate the reviewer’s insightful comment. We have moved this sentence from the result section to the method section as indicated in line 278-279 in the revised manuscript.

4. Please definite the units of longitudinal coordinates in Figure 2C, Figure 3C, Figure 3D, Figure 4 and Figure 5.

Response: We appreciate the reviewer’s insightful comment. We have revised the units of longitudinal coordinates in each figure as suggested by the reviewer in this revised manuscript.

Section Editor:

All the reviewers agree that the data presented on the differential transcriptional expression signatures of alveolar macrophages in mice infected with Cryptococcus neoformans and C. gattii are novel and potentially translatable. The reviewers also believe that the experiments are well-designed and the results are clearly presented and has significant impact to the field.

• However, each reviewer has raised specific concerns that need to be addressed. Please respond to these comments in a timely manner.

In addition to the reviewers' feedback, I have one minor comment regarding Figure 1. As currently shown, it appears that the mice were infected with both C. neoformans and C. gattii. To improve clarity, consider adding the word “or” between the two yeast strains in the figure to emphasize that the infection was with one strain or the other—not a mixture of both.

Response: We appreciate editor’s comment. In the revised manuscript, we have added the word “or” between the two yeast strains in Fig. 1A.

---

## [Decision Letter · Decision Letter 1]

3 Sep 2025

PONE-D-25-29534R1Transcriptomic profiling of lung alveolar macrophages reveals distinct contribution of sterol metabolism in macrophage response to Cryptococcus gattii infectionPLOS ONE

Dear Dr., Angkasekwinai,

Thank you for revising your manuscript and addressing all the comments of the reviewers. Both of the reviewers are satisfied with the revision and the have recommended for the acceptance of the manuscript after very minor comments. Please review the manuscript one more time just to make sure that there no typos or italicization of the names etc. 

Please submit your revised manuscript at your earliest. Please include the following items when submitting your revised manuscript:

We look forward to receiving your revised manuscript.

Kind regards,

Rajendra Upadhya

Academic Editor

PLOS ONE

Journal Requirements:

Reviewers' comments:

Reviewer's Responses to Questions

**Comments to the Author**

1. If the authors have adequately addressed your comments raised in a previous round of review and you feel that this manuscript is now acceptable for publication, you may indicate that here to bypass the “Comments to the Author” section, enter your conflict of interest statement in the “Confidential to Editor” section, and submit your "Accept" recommendation.

Reviewer #1: All comments have been addressed

Reviewer #2: All comments have been addressed

2. Is the manuscript technically sound, and do the data support the conclusions?

Reviewer #1: Yes

Reviewer #2: Yes

3. Has the statistical analysis been performed appropriately and rigorously? 

Reviewer #1: Yes

Reviewer #2: Yes

4. Have the authors made all data underlying the findings in their manuscript fully available?

Reviewer #1: Yes

Reviewer #2: Yes

5. Is the manuscript presented in an intelligible fashion and written in standard English?

Reviewer #1: Yes

Reviewer #2: Yes

6. Review Comments to the Author

Reviewer #1: The authors have well addressed my previous concerns and verified them with sufficient supplementary experiments. I agree to publish this study in its current form

Reviewer #2: The manuscript has been edited and is being improved.

I suggest a quick look at the text for spelling corrections, including italicizing words where necessary, for example.

Authors should correct the text on lines 54-56.

Where it says: A hallmark of both species is the presence of a polysaccharide capsule composed of glucuronoxylomannan (GXM), galactoxylomannan (GalXM), and glucuronoxylomanogalactan (GXMGal). It should be changed to: A hallmark of both species is the presence of a polysaccharide capsule composed of glucuronoxylomannan (GXM) and glucuronoxylomanogalactan (GXMGal).

7. PLOS authors have the option to publish the peer review history of their article (what does this mean? ). If published, this will include your full peer review and any attached files.

**Do you want your identity to be public for this peer review?** For information about this choice, including consent withdrawal, please see our Privacy Policy .

Reviewer #1: No

Reviewer #2: No

---

## [Author Response · Author response to Decision Letter 2]

5 Sep 2025

Response to Reviewers

Reviewer 2: The manuscript has been edited and is being improved.

I suggest a quick look at the text for spelling corrections, including italicizing words where necessary, for example.

Authors should correct the text on lines 54-56.

Where it says: A hallmark of both species is the presence of a polysaccharide capsule composed of glucuronoxylomannan (GXM), galactoxylomannan (GalXM), and glucuronoxylomanogalactan (GXMGal). It should be changed to: A hallmark of both species is the presence of a polysaccharide capsule composed of glucuronoxylomannan (GXM) and glucuronoxylomanogalactan (GXMGal).

Response: We appreciate the reviewer’s thoughtful suggestion. We have revised the paragraph in the introduction to this revised manuscript (lines 54–55).

Section Editor:

Thank you for revising your manuscript and addressing all the comments of the reviewers. Both of the reviewers are satisfied with the revision and the have recommended for the acceptance of the manuscript after very minor comments. Please review the manuscript one more time just to make sure that there no typos or italicization of the names etc.

Response: Thank you very much for your message and for handling our manuscript. We truly appreciate the constructive feedback and recommendation for acceptance provided by the reviewers. We have carefully reviewed the manuscript to ensure that all remaining typos, formatting issues, or italicization of names have been corrected before resubmission. Additionally, we thoroughly checked all cited references to confirm their completeness and accuracy, and to ensure that no retracted papers have been included. Furthermore, we have uploaded all figure files to the PACE digital diagnostic tool and download the suggested version to ensure that figures fully comply with PLOS formatting requirements.

---

## [Editor Report · Decision Letter 2]

10 Sep 2025

Transcriptomic profiling of lung alveolar macrophages reveals distinct contribution of sterol metabolism in macrophage response to Cryptococcus gattii infection

PONE-D-25-29534R2

Dear Dr. Angkasekwinai 

We’re pleased to inform you that your manuscript has been judged scientifically suitable for publication and will be formally accepted for publication once it meets all outstanding technical requirements.

Kind regards,

Rajendra Upadhya

Academic Editor

PLOS ONE
---

## [Editor Report · Acceptance letter]

PONE-D-25-29534R2

PLOS ONE

Dear Dr. Angkasekwinai,

I'm pleased to inform you that your manuscript has been deemed suitable for publication in PLOS ONE. Congratulations! Your manuscript is now being handed over to our production team.

Kind regards,

on behalf of

Dr. Rajendra Upadhya

Academic Editor

PLOS ONE